# Dynamical modeling of nonlinear latent factors in multiscale neural activity with real-time inference

**Eray Erturk**
Ming Hsieh Department of Electrical and Computer Engineering
University of Southern California
Los Angeles, CA
eerturk@usc.edu

**Maryam M. Shanechi**[*]
Ming Hsieh Department of Electrical and Computer Engineering
Thomas Lord Department of Computer Science
Alfred E. Mann Department of Biomedical Engineering
Neuroscience Graduate Program
University of Southern California
Los Angeles, CA
shanechi@usc.edu

## Abstract

Real-time decoding of target variables from multiple simultaneously recorded neural time-series modalities, such as discrete spiking activity and continuous field potentials, is important across various neuroscience applications. However, a major challenge for doing so is that different neural modalities can have different timescales (i.e., sampling rates) and different probabilistic distributions, or can even be missing at some time-steps. Existing nonlinear models of multimodal neural activity do not address different timescales or missing samples across modalities. Further, some of these models do not allow for real-time decoding. Here, we develop a learning framework that can enable real-time recursive decoding while nonlinearly aggregating information across multiple modalities with different timescales and distributions and with missing samples. This framework consists of 1) a multiscale encoder that nonlinearly aggregates information after learning within-modality dynamics to handle different timescales and missing samples in real time, 2) a multiscale dynamical backbone that extracts multimodal temporal dynamics and enables real-time recursive decoding, and 3) modality-specific decoders to account for different probabilistic distributions across modalities. In both simulations and three distinct multiscale brain datasets, we show that our model can aggregate information across modalities with different timescales and distributions and missing samples to improve real-time target decoding. Further, our method outperforms various linear and nonlinear multimodal benchmarks in doing so.

## 1 Introduction

Real-time continuous decoding of target time-series, such as various brain or behavioral states from neural time-series data is of interest across many neuroscience applications. A popular approach for doing so is to develop dynamical latent factor models that describe neural dynamics in terms

---

[*]Corresponding author: shanechi@usc.edu

39th Conference on Neural Information Processing Systems (NeurIPS 2025).

of the temporal evolution of latent variables that can be used for downstream decoding. To date, dynamical latent factor models of neural data have mostly focused on a single modality of neural data, for example, either spiking activity or local field potentials (LFP) [1–4]. However, brain and behavioral target states are encoded across multiple spatial and temporal scales of brain activity that are measured with different neural modalities. Furthermore, some of these dynamical models have a non-causal inference procedure, which hinders real-time decoding. Therefore, inference of target variables could be improved by developing nonlinear dynamical models of multimodal neural time-series that can, at each time-step, aggregate information across neural modalities and do so in real-time.

A natural challenge in developing such multimodal models arises when modalities are not aligned due to their different recording timescales that can be caused by various factors such as fundamental biological differences across modalities—with some modalities evolving slower than others [5]—differences in recording devices [6, 7], or measurement failures or interruptions [8–10]. Further, modalities could have different distributions. For example, spiking activity is a binary-valued time-series that indicates the presence of action potential events from neurons at each time. As such, it has a fast millisecond timescale and is often modeled as count processes, such as Poisson. In comparison, LFP activity is a continuous-valued modality that measures network-level neural processes, has a slower timescale, and is typically modeled with a Gaussian distribution [5, 7, 11]. We refer to multimodal data with different timescales as **multiscale data**. Thus, to fuse information across spiking and LFP modalities and improve downstream target decoding tasks, their dynamics should be modeled by incorporating their cross-modality probabilistic and timescale differences through a careful encoder design.

Existing neural dynamical modeling approaches do not address the nonlinear modeling of multimodal data with different timescales and/or with real-time decoding capability. Specifically, most dynamical models do not capture multimodal neural dynamics and instead focus on a single modality of neural activity either by using linear/switching-linear approaches [1, 12, 13] or by utilizing nonlinear deep learning approaches [2–4, 14]. There are also some methods to jointly model unimodal neural activity together with behavior [15–23], but again, their latent factor inference is performed by processing unimodal neural time-series and does not aggregate multimodal neural data. While there has been some recent work on dynamical modeling and decoding of multimodal neural data, many of these works have been linear [24–28]. Motivated by this gap, recent studies have developed nonlinear models of multimodal neural data [18, 29–31]. However, these recent works have not addressed modalities with different timescales; further, the latent factor inference in such dynamical models has been done non-causally over time [29–31].

Beyond neural time-series data, many approaches in other domains have been proposed to combine multiple modalities. However, these are again not focused on addressing the challenge of different timescales and missing samples over time, and their applicability to joint modeling of Poisson and Gaussian distributed modalities encountered in neuroscience has not been investigated (Section 2).

**Contributions** We introduce **M**ultiscale **R**eal-time **I**nference of **N**onlinear **E**mbeddings (MRINE), a nonlinear dynamical modeling approach designed to nonlinearly fuse multimodal neural time-series with different timescales, distinct distributions, and/or missing samples over time, while supporting inference both in real-time and non-causally. To achieve these capabilities, we: **1)** design a multiscale encoder that performs nonlinear information fusion via neural networks after learning modality-specific dynamical models that account for timescale differences and missing samples in real-time by learning the temporal dynamics (Section 3.2) and **2)** impose smoothness priors on the latent dynamics via smoothness regularization objectives that also prevent learning trivial identity neural network transformations (Section 3.3).

Through stochastic Lorenz attractor simulations and two independent nonhuman primate (NHP) spiking and LFP neural datasets, we show that MRINE infers latent factors that are more predictive of target variables such as the NHP's arm kinematics. Further, we compare MRINE with various recent linear and nonlinear multimodal methods and show that MRINE outperforms them in downstream decoding from multiscale spike-LFP time-series in the NHP datasets, as well as on a high-dimensional visual stimuli dataset containing neuropixel spikes and calcium imaging data.

## 2 Related work

**Single-Scale Models of Neural Activity**  Numerous dynamical models of single-scale neural activity have been developed. Some of these models are in linear or generalized linear form [1, 12, 15, 32–34]. Linear dynamical models (LDMs) are widely used in real-time applications because they provide real-time and recursive inference algorithms. To enable nonlinear modeling, there has been an increased interest in switching linear models [13] and deep learning architectures including recurrent neural networks (RNN) with nonlinear temporal dynamics [3, 19, 35], autoencoder-based architectures that utilize Markovian linear dynamics to learn a smoothing distribution [2, 14, 36], transformer encoder based models optimized with masked training [37, 38] and neural ordinary differential equations [39]. These models have shown great promise in improving behavior decoding compared to linear models [3]. A recent work [4] has also developed a nonlinear neural network framework termed DFINE that supports flexible inference—i.e., enables both real-time filtering and noncausal smoothing, and accounts for missing observations simultaneously—by jointly training an autoencoder with linear state-space models and utilizing Kalman filtering [40]. Another work has proposed a low-rank structured variational encoding framework for Gaussian state-space models to capture dense covariances with predictive capabilities, while supporting real-time parameterization of their inference network [41]. While LDM-based approaches and some other nonlinear approaches [41] can handle missing samples-—either via Kalman filtering or in a similar spirit within probabilistic state-space formulations—-all the above linear and nonlinear methods are designed for a single modality of neural activity and do not address multiscale modeling.

**Multimodal Information Fusion**  Outside neuroscience, fusing multiple data modalities has been extensively researched across many areas including natural language processing (NLP) and computer vision. However, these approaches do not address nonlinear modeling of multiple time-series modalities with different timescales and with real-time inference capability, which we consider here. Specifically, in computer vision, many studies focused on variational autoencoders (VAE), and approximated the joint posterior distribution by factorization it into modality-specific posteriors [42–46] to handle missing modalities, or by concatenating the modality-specific representations [47, 48]. Instead of learning the common embedding space via factorization or concatenation, some studies have also employed cross-modality generation [49, 50]. Even though some of these methods can fuse time-series modalities with different timescales, they need to do so using separate networks for each modality to first non-causally encode each modality into a single vector, which means information fusion cannot be done causally over time-steps (i.e., in real time). However, many neuroscience applications require aggregating information at each time step in a causal manner to perform continuous real-time decoding of target variables. Finally, they are not designed to handle time-series signals with missing samples in time and would rely on data augmentations such as zero-padding, which can be suboptimal by changing the value of missing samples [51–53].

**Multimodal Models in Neuroscience**  One line of work in neuroscience aims to jointly model single-scale neural activity and behavior as multimodal signals [15–17, 19]. However, in these works, latent factor inference is performed using single-scale neural activity without any information fusion, similar to the single-scale models discussed above. Another line of work aims to model multimodal neural signals. However, these methods are either linear/generalized-linear or are designed for offline reconstruction without addressing distinct timescales or enabling real-time inference. Specifically, some approaches proposed linear multimodal modeling frameworks [26, 27] and learned the model parameters via expectation-maximization (EM). But these methods do not capture nonlinearity and further do not address different timescales. To partly address this gap, a linear dynamical modeling framework was introduced in [24] in which model parameters are also learned with EM but this time inference can aggregate modalities with different timescales. With a similar linear formulation of multiscale dynamics, recent work in [28] proposed a more computationally efficient learning framework compared to [24] by using subspace identification. However, both of these approaches are still linear and cannot characterize nonlinearities.

To account for nonlinearity, recent studies have developed nonlinear latent factor models for multi-modal neural data [18, 29–31], but their formulation assumes the same timescale for the different modalities and also does not consider missing samples. Thus, in such situations, they would need to rely on indirect approaches such as augmentations with zero-padding, which can be suboptimal by changing the value of missing samples [4, 51–53]. Furthermore, such multimodal dynamical models have non-causal inference networks and thus do not enable real-time inference of latent

factors [29–31]. Instead, here we develop a nonlinear dynamical modeling framework for multimodal neural time-series data that supports real-time and efficient recursive inference, and handles both timescale differences and missing samples by directly leveraging the learned dynamical model to predict these missing samples.

## 3   Methodology

We assume that we observe discrete neural signals (e.g., spikes) $\boldsymbol{s}_t \in \{0,1\}^{n_s}$ for $t \in \mathcal{T}$ where $\mathcal{T} = \{1, 2, \ldots, T\}$ and continuous neural signals (e.g., LFPs) $\boldsymbol{y}_{t'} \in \mathbb{R}^{n_y}$ for $t' \in \mathcal{T}'$ where $\mathcal{T}' \subseteq \mathcal{T}$. Note that the two different sets $\mathcal{T}$ and $\mathcal{T}'$ allow for the timescale differences of $\boldsymbol{s}_t$ and $\boldsymbol{y}_{t'}$ via different time-indices. As shown in Fig. 1 and expanded on below, we describe the neural processes generating $\boldsymbol{s}_t$ and $\boldsymbol{y}_{t'}$ through multiscale latent and embedding factors, which in turn can be extracted by nonlinearly aggregating information across these multiple neural modalities with a multiscale encoder.

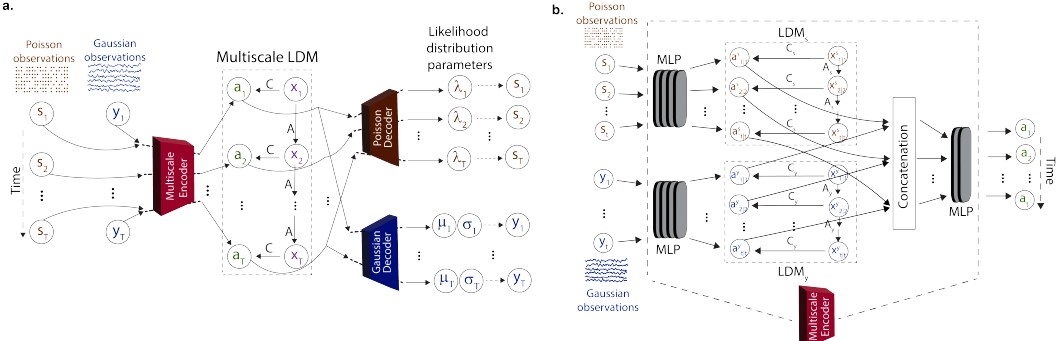

Figure 1: **a.** MRINE model architecture. Multiscale encoder nonlinearly extracts multiscale embedding factors ($\boldsymbol{a}_t$) by fusing discrete Poisson and continuous Gaussian neural time-series in real-time (as indicated by thick dashed lines) while accounting for timescale differences and missing samples. Temporal dynamics are then explained with a multiscale LDM whose states are the multiscale latent factors ($\boldsymbol{x}_t$). As an example, when $\boldsymbol{y}_2$ is missing, $\boldsymbol{a}_2$ and $\boldsymbol{x}_2$ are inferred by processing only $\boldsymbol{s}_2$. **b.** MRINE multiscale encoder design. Modality-specific LDMs learn within-modality dynamics and account for timescale differences or missing samples via Kalman filtering. Then, filtered modality-specific embedding factors ($\boldsymbol{a}_{t|t}^s$ and $\boldsymbol{a}_{t|t}^y$) are fused and processed by another fusion network to obtain the multiscale embedding factors $\boldsymbol{a}_t$. As an example, when $\boldsymbol{y}_2$ is missing, $\boldsymbol{a}_{2|2}^y$ is predicted only by the dynamics of modality-specific LDM in Eq. 7. Note that MLP networks are time-invariant, and information about temporal dynamics is incorporated through LDMs.

### 3.1   Model formulation

The model architecture is shown in Fig. 1a with its main components being a multiscale encoder network for nonlinear information fusion, a multiscale LDM backbone to facilitate real-time inference, and nonlinear decoder networks for each modality. We write the generative model as follows:

$$\boldsymbol{x}_{t+1} = \boldsymbol{A}\boldsymbol{x}_t + \boldsymbol{w}_t \tag{1}$$

$$\boldsymbol{a}_t = \boldsymbol{C}\boldsymbol{x}_t + \boldsymbol{r}_t \tag{2}$$

$$p_{\theta_s}(\boldsymbol{s}_t \mid \boldsymbol{a}_t) = \text{Poisson}(\boldsymbol{s}_t; \boldsymbol{\lambda}(\boldsymbol{a}_t)), \\ \text{where } \boldsymbol{\lambda}(\cdot) := f_{\theta_s}(\cdot) \tag{3}$$

$$p_{\theta_y}(\boldsymbol{y}_{t'} \mid \boldsymbol{a}_{t'}) = \mathcal{N}(\boldsymbol{y}_{t'}; \boldsymbol{\mu}(\boldsymbol{a}_{t'}), \boldsymbol{\sigma}), \\ \text{where } \boldsymbol{\mu}(\cdot) := f_{\theta_y}(\cdot) \tag{4}$$

Here, Eq. 1 and 2 form a multiscale LDM, and $t \in \mathcal{T}$ where $\mathcal{T} = \{1, 2, \ldots, T\}$ and $t' \in \mathcal{T}'$ where $\mathcal{T}' \subseteq \mathcal{T}$. In this LDM, $\boldsymbol{a}_t \in \mathbb{R}^{n_a}$, termed multiscale embedding factors, are the observations. We obtain $\boldsymbol{a}_t$ as the nonlinear aggregation of multimodal information through the multiscale encoder

designed in Section 3.2. Further, $\boldsymbol{x}_t \in \mathbb{R}^{n_x}$, termed multiscale latent factors, are the LDM states and model the dynamics as linear in the nonlinear embedding space, which facilitates real-time inference (Section 3.2). Note that from this point onward, we will only use time-index $t$ (except observation models) for embedding and latent factors as $\mathcal{T}' \subseteq \mathcal{T}$. Correspondingly, $\boldsymbol{A} \in \mathbb{R}^{n_x \times n_x}$ and $\boldsymbol{C} \in \mathbb{R}^{n_a \times n_x}$ are the state transition and observation matrices in the multiscale LDM, respectively; $\boldsymbol{w}_t \in \mathbb{R}^{n_x}$ and $\boldsymbol{r}_t \in \mathbb{R}^{n_a}$ are zero-mean Gaussian dynamics and observation noises with covariance matrices $\boldsymbol{W} \in \mathbb{R}^{n_x \times n_x}$ and $\boldsymbol{R} \in \mathbb{R}^{n_a \times n_a}$.

The observations from different modalities $\boldsymbol{s}_t$ and $\boldsymbol{y}_{t'}$ are then generated from $\boldsymbol{a}_t$ (or $\boldsymbol{a}_{t'}$) as in Eqs. 3 and 4, respectively, with the likelihood distributions denoted by $p_{\theta_s}(\boldsymbol{s}_t \mid \boldsymbol{a}_t)$ and $p_{\theta_y}(\boldsymbol{y}_{t'} \mid \boldsymbol{a}_{t'})$. Assuming that data modalities are conditionally independent given $\boldsymbol{a}_t$ (or $\boldsymbol{a}_{t'}$ for $\boldsymbol{y}_{t'}$), we modeled the discrete spiking activity $\boldsymbol{s}_t$ with a Poisson distribution and the continuous neural signals $\boldsymbol{y}_{t'}$ with a Gaussian distribution, given that these distributions have shown success in modeling each of these modalities [2, 3, 24, 54]. The means of the corresponding distributions $\boldsymbol{\lambda}(\cdot) := f_{\theta_s}(\cdot) \in \mathbb{R}^{n_s}$ and $\boldsymbol{\mu}(\cdot) := f_{\theta_y}(\cdot) \in \mathbb{R}^{n_y}$ are parametrized by neural networks with parameters $\theta_s$ and $\theta_y$. Practically, we observed that learning the variance of the Gaussian likelihood yielded suboptimal performance, thus we set it to a constant value, i.e., unit variance, as in previous works [55, 56].

## 3.2 Encoder design and inferring multiscale factors

To infer $\boldsymbol{a}_t$ and $\boldsymbol{x}_t$ from $\boldsymbol{s}_t$ and $\boldsymbol{y}_{t'}$, we first construct the mapping from $\boldsymbol{s}_t$ and $\boldsymbol{y}_{t'}$ to $\boldsymbol{a}_t$ as:

$$\boldsymbol{a}_t = f_\phi(\boldsymbol{s}_t, \boldsymbol{y}_{t'}) \tag{5}$$

where $f_\phi(\cdot)$ represents the multiscale encoder network parametrized by a neural network with parameters $\phi$.

One obstacle in the design of the encoder network is accounting for the different timescales without using augmentation techniques such as zero-padding as commonly done [53, 57] since they can yield suboptimal performance and distort the information during latent factor inference [51, 52, 58]. Thus, it is important to account for timescale differences and the possibility of missing samples when designing the multiscale encoder. Additionally, our goal is to perform multimodal fusion at each time-step while also allowing for real-time inference of factors. We address these problems, which remain unresolved in prior methods, with a multiscale encoder network design shown in Fig. 1b.

In our multiscale encoder (Fig. 1b), at each time-step, first, each modality ($\boldsymbol{s}_t$ and $\boldsymbol{y}_{t'}$) is processed by separate time-invariant multilayer perception (MLP) networks with parameters $\phi_s$ and $\phi_y$ to obtain modality-specific embedding factors, $\boldsymbol{a}_t^s \in \mathbb{R}^{n_a}$ and $\boldsymbol{a}_t^y \in \mathbb{R}^{n_a}$, respectively. Then, we construct modality-specific LDMs for each modality, whose observations at each time-step are their corresponding embedding factors:

$$\begin{aligned} \boldsymbol{x}_{t+1}^s &= \boldsymbol{A}_s \boldsymbol{x}_t^s + \boldsymbol{w}_t^s \\ \boldsymbol{a}_t^s &= \boldsymbol{C}_s \boldsymbol{x}_t^s + \boldsymbol{r}_t^s \end{aligned} \tag{6}$$

$$\begin{aligned} \boldsymbol{x}_{t+1}^y &= \boldsymbol{A}_y \boldsymbol{x}_t^y + \boldsymbol{w}_t^y \\ \boldsymbol{a}_t^y &= \boldsymbol{C}_y \boldsymbol{x}_t^y + \boldsymbol{r}_t^y \end{aligned} \tag{7}$$

where $\boldsymbol{x}_t^s, \boldsymbol{x}_t^y \in \mathbb{R}^{n_x}$ are modality-specific latent factors, $\boldsymbol{A}_s, \boldsymbol{A}_y \in \mathbb{R}^{n_a \times n_a}$ are the state transition matrices, $\boldsymbol{C}_s, \boldsymbol{C}_y \in \mathbb{R}^{n_a \times n_a}$ are the emission matrices, $\boldsymbol{w}_t^s, \boldsymbol{w}_t^y \in \mathbb{R}^{n_a}$ are the zero-mean dynamics noises with covariances $\boldsymbol{W}_s, \boldsymbol{W}_y \in \mathbb{R}^{n_a \times n_a}$, and $\boldsymbol{r}_t^s$ and $\boldsymbol{r}_t^y$ are the zero-mean Gaussian observation noises with covariances $\boldsymbol{R}_s, \boldsymbol{R}_y \in \mathbb{R}^{n_a \times n_a}$ for modalities $\boldsymbol{s}_t$ and $\boldsymbol{y}_{t'}$, respectively. We denote the modality-specific LDM parameters by $\psi_s = \{\boldsymbol{A}_s, \boldsymbol{C}_s, \boldsymbol{W}_s, \boldsymbol{R}_s\}$ and $\psi_y = \{\boldsymbol{A}_y, \boldsymbol{C}_y, \boldsymbol{W}_y, \boldsymbol{R}_y\}$ for $\boldsymbol{s}_t$ and $\boldsymbol{y}_{t'}$, respectively.

In our design, we use the modality-specific LDMs because they allow us to account for missing samples (whether due to timescale differences or missed measurements) by using the learned within-modality state dynamics to predict these samples forward in time, while also maintaining the operation fully real-time/causal. Specifically, given the modality-specific LDMs in Eq. 6 and 7, we can obtain the modality-specific latent factors, $\boldsymbol{x}_{t|t}^s$ and $\boldsymbol{x}_{t|t}^y$ with Kalman filtering, which is real-time and constitutes the optimal minimum mean-squared error estimator for these models [40]. We use the subscript $i|j$ to denote the factors inferred at time $i$ given all observations up to time $j$. As such, subscripts $t|t$, $t|T$ and $t+k|t$ denote causal/real-time filtering, non-causal smoothing and $k$-step-ahead prediction, respectively. At this stage, if $t$ is an intermittent time-step such that $\boldsymbol{y}_t$ is missing (i.e., $t \in \mathcal{T}$ and $t \notin \mathcal{T}'$), $\boldsymbol{x}_{t|t}^y$ is obtained with forward prediction using the Kalman

predictor as $\boldsymbol{x}_{t|t}^y = \boldsymbol{A}_y \boldsymbol{x}_{t-1|t-1}^y$ [59], and similarly for $\boldsymbol{x}_{t|t}^s$ if $\boldsymbol{s}_t$ is missing. Having done this for each modality, we perform information fusion in real-time by concatenating the modality-specific embedding factors and passing them through a fusion network with parameters $\phi_m$ to obtain the initial representation for $\boldsymbol{a}_t$, which later becomes the noisy observations of the multiscale LDM formed by Eq. 1 and 2 (also see Fig. 1a). We denote the learnable multiscale encoder network parameters by $\phi = \{\phi_s, \phi_y, \psi_s, \psi_y, \phi_m\}$.

The multiscale LDM now allows us to infer $\boldsymbol{x}_{t|t}$ with real-time (causal) Kalman filtering, or infer $\boldsymbol{x}_{t|T}$ with non-causal Kalman smoothing [60]. Similarly, $k$-step-ahead predicted multiscale latent factors $\boldsymbol{x}_{t+k|t}$ can be obtained by forward propagating $\boldsymbol{x}_{t|t}$ $k$-times into the future with Eq. 1, i.e., $\boldsymbol{x}_{t+k|t} = \boldsymbol{A}^k \boldsymbol{x}_{t|t}$. We denote the parameters of the multiscale LDM by $\psi_m = \{\boldsymbol{A}, \boldsymbol{C}, \boldsymbol{W}, \boldsymbol{R}\}$. We can now obtain the filtered, smoothed, and $k$-step-ahead predicted parameters of the likelihood functions in Eq. 3 and 4 by first using Eq. 2 to compute the corresponding multiscale embedding factors—i.e. $\boldsymbol{a}_{i|j} = \boldsymbol{C}\boldsymbol{x}_{i|j}$, where $i|j$ is $t|t$, $t|T$, and $t+k|t$, respectively—and then forward passing these factors through $f_{\theta_s}(\cdot)$ and $f_{\theta_y}(\cdot)$ (Fig. 1a).

### 3.3 Learning the model parameters

$k$-**step ahead prediction** To learn the MRINE model parameters and in order to encourage learning the dynamics, as part of the loss, we employ the multi-horizon $k$-step-ahead prediction loss defined as:

$$\mathcal{L}_k = -\sum_{k \in \mathcal{K}} \Big( \sum_{\substack{t \in \mathcal{T} \\ t \geq k}} \tau \log\left(p_{\theta_s}(\boldsymbol{s}_t \mid \boldsymbol{a}_{t|t-k})\right) + \sum_{\substack{t' \in \mathcal{T}' \\ t' \geq k}} \log\left(p_{\theta_y}(\boldsymbol{y}_{t'} \mid \boldsymbol{a}_{t'|t'-k})\right) \Big) \tag{8}$$

where $\mathcal{T}$ and $\mathcal{T}'$ denote the time-steps when $\boldsymbol{s}_t$ and $\boldsymbol{y}_{t'}$ are observed, respectively. $\tau$ is the scaling parameter as the log-likelihood values of different modalities are of different scales (see Appendix A.2.2), and $\mathcal{K}$ is the set of future prediction horizons. We note that $k$-step-ahead prediction is performed by computing $k$-step-ahead predicted multiscale latent factors, $\boldsymbol{x}_{t+k|t}$, rather than modality-specific ones.

**Smoothed reconstruction** In addition to the $k$-step-ahead prediction, we also optimize the reconstruction from smoothed multiscale factors:

$$\mathcal{L}_{smooth} = -\Big( \sum_{t \in \mathcal{T}} \tau \log\left(p_{\theta_s}(\boldsymbol{s}_t \mid \boldsymbol{a}_{t|T})\right) + \sum_{t' \in \mathcal{T}'} \log\left(p_{\theta_y}(\boldsymbol{y}_{t'} \mid \boldsymbol{a}_{t'|T})\right) \Big) \tag{9}$$

where $T$ is the last time-step that any modality is observed.

**Smoothness regularization** To impose a smoothness prior on learned dynamics and to prevent the model from overfitting by learning trivial identity encoder/decoder transformations, in our loss, we also apply smoothness regularization on $p_{\theta_s}(\boldsymbol{s}_t \mid \boldsymbol{a}_{1:T})$, $p_{\theta_y}(\boldsymbol{y}_{t'} \mid \boldsymbol{a}_{1:T})$ and $p(\boldsymbol{x}_t \mid \boldsymbol{a}_{1:T})$ by minimizing the KL-divergence between the distributions in consecutive time-steps as introduced in [61] for Gaussian-distributed modalities. Here, we extend this technique also to Poisson-distributed modalities as below. Let $\mathcal{L}_{sm}$ be the smoothness regularization penalty, defined as:

$$\mathcal{L}_{sm} = \gamma_s \underbrace{\sum_{i=1}^{|\mathcal{T}|-1} \sum_{j=1}^{n_s} d\Big(p_{\theta_s}(s_{\mathcal{T}_i}^j | \boldsymbol{a}_{\mathcal{T}_i|T}), p_{\theta_s}(s_{\mathcal{T}_{i+1}}^j | \boldsymbol{a}_{\mathcal{T}_{i+1}|T})\Big)}_{\text{Smoothness on } \boldsymbol{s}_t} + \gamma_y \underbrace{\sum_{i=1}^{|\mathcal{T}'|-1} \sum_{j=1}^{n_y} d\Big(p_{\theta_y}(y_{\mathcal{T}_i'}^j | \boldsymbol{a}_{\mathcal{T}_i'|T}), p_{\theta_y}(y_{\mathcal{T}_{i+1}'}^j | \boldsymbol{a}_{\mathcal{T}_{i+1}'|T})\Big)}_{\text{Smoothness on } \boldsymbol{y}_{t'}}$$
$$+ \gamma_x \underbrace{\sum_{t=1}^{T} \sum_{j=1}^{\lfloor \frac{n_x}{2} \rfloor} d\Big(p(x_t^j | \boldsymbol{a}_{t|T}), p(x_{t+1}^j | \boldsymbol{a}_{t+1|T})\Big)}_{\text{Smoothness on } \boldsymbol{x}_t} \tag{10}$$

where $d(\cdot, \cdot)$ is the KL-divergence between given distributions, subscript $i$ denotes the $i^{\text{th}}$ element of the set, superscript $j$ is the $j^{\text{th}}$ component of the vector (e.g., $\boldsymbol{s}_{\mathcal{T}_i}$), and $p_{\theta_s}(\boldsymbol{s}_t \mid \boldsymbol{a}_{t|T})$ and $p_{\theta_y}(\boldsymbol{y}_{t'} \mid \boldsymbol{a}_{t'|T})$ are as in Eq. 3 and 4, respectively. Here $\gamma_s$, $\gamma_y$ and $\gamma_x$ are the scaling hyperparameters. The smoothness penalties on $\boldsymbol{s}_t$ and $\boldsymbol{y}_{t'}$ are computed over the time-steps that they are observed. The penalty on $\boldsymbol{x}_t$ is obtained over all time-steps as $\boldsymbol{x}_t$ is inferred for all time-steps. After extracting $\boldsymbol{a}_{1:T}$ with multiscale encoder, $p(\boldsymbol{x}_t \mid \boldsymbol{a}_{1:T})$ can be obtained with the Kalman smoother, which provides the posterior distribution for the multiscale LDM [40, 62]:

$$p(\boldsymbol{x}_t \mid \boldsymbol{a}_{1:T}) = \mathcal{N}(\boldsymbol{x}_t; \boldsymbol{x}_{t|T}, \Sigma_{t|T}) \tag{11}$$

where $\boldsymbol{x}_{t|T}$ and $\Sigma_{t|T}$ are the smoothed multiscale latent factors and their error covariances, respectively. To allow the model to learn both fast and slow dynamics, we put the smoothness regularization on $\boldsymbol{x}_t$ on half of its dimensions.

To assess the impact of incorporating smoothness regularization terms in Eq. 10 and smoothed reconstruction in Eq. 9, we performed an ablation study (see Appendix A.6.2), which demonstrates that each term contributes to the improved performance. Further, in another ablation study on the effect of multiscale modeling (see Appendix A.6.4), we show that MRINE's multiscale encoder design is an important contributing factor to improved performance, compared to the case where missing samples are imputed by zeros and removed from the training objectives above.

Finally, we form the loss as the sum of the above elements and regularization terms, and minimize it via mini-batch gradient descent using the Adam optimizer [63] to learn the model parameters $\{\phi, \psi, \theta_s, \theta_y\}$:

$$\mathcal{L}_{MRINE} = \mathcal{L}_k + \mathcal{L}_{smooth} + \mathcal{L}_{sm} + \gamma_r L_2(\theta_s, \theta_y, \phi_s, \phi_y, \phi_m) \tag{12}$$

where $L_2(\cdot)$ is the $L_2$ regularization penalty on the MLP weights and $\gamma_r$ is the scaling hyperparameter.

Moreover, we employ a dropout technique termed *time-dropout* during training to increase the robustness of MRINE to missing samples even further. See Appendix A.1 for more information, Appendix A.6.1 for an ablation study on the effect of *time-dropout*, and Appendix A.2 for training details and hyperparameters.

## 4  Results

### 4.1  Stochastic Lorenz attractor simulations

We first validated that MRINE can successfully aggregate information across multiple modalities by performing simulations with the stochastic Lorenz attractor dynamics defined in Eq. 18. To do so, we generated Poisson and Gaussian observations with 5, 10 and 20 dimensions as described in Appendix A.4.2. We generated 4 systems with different random seeds and performed 5-fold cross-validation for each system. Then, we trained MRINE as well as single-scale networks with either only Gaussian observations or only Poisson observations (see Appendix A.2.1). To assess MRINE's ability to aggregate multimodal information, we compared its latent reconstruction accuracies with those of single-scale networks. For each model, these accuracies were obtained by computing the average correlation coefficient (CC) between the true and reconstructed latent factors (see Appendix A.4.3 for details). We refer to each observation dimension as a channel for simplicity.

We first considered the Poisson modality as the primary modality and fused with it 5, 10, and 20 Gaussian channels as the secondary modality. As shown in Fig. 2a, for all different numbers of Poisson channels, gradually fusing Gaussian channels significantly improved the latent reconstruction accuracies ($p < 10^{-5}$, $n = 20$, one-sided Wilcoxon signed-rank test). As expected, these improvements were higher in the low information regime where fewer primary Poisson channels were available (5 and 10) compared to the high information regime (20 Poisson channels). Likewise, when the Gaussian modality was the primary modality, latent reconstruction accuracies showed consistent improve-

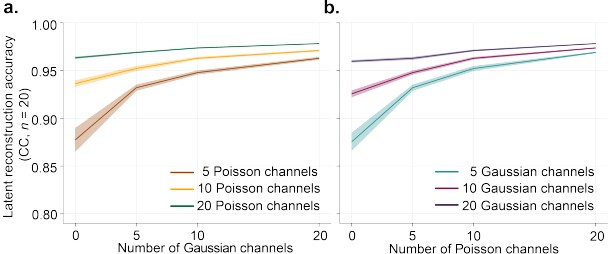

Figure 2: Latent reconstruction accuracies for the stochastic Lorenz attractor simulations. **a.** Accuracies when 5, 10, or 20 Poisson channels were the primary modality to which Gaussian channels were gradually added. Solid lines show the mean and shaded areas show the standard error of the mean (SEM). **b.** Similar to **a**, but with the Gaussian channels being the primary modality.

ment across all regimes as Poisson channels were fused (Fig. 2b, $p < 10^{-5}$, $n = 20$, one-sided Wilcoxon signed-rank test).

## 4.2 MRINE fused multiscale information in behavior decoding for the NHP datasets

To test MRINE's information aggregation capabilities in real datasets, we used two independent publicly available NHP datasets [64–66]. For both datasets, discrete spiking activity and continuous LFP signals were recorded while subjects performed 2D reaches either on a grid via a cursor (grid-reaching dataset), or from a center target to a random outer target via a manipulandum (center-out reaching dataset). We considered the 2D cursor and manipulandum velocities in x and y directions as our target behavior variables (see Appendices A.5.1 and A.5.2 for details). For both datasets, we trained single-scale models with 5, 10, and 20 channels of either spike or LFP signals alone, as well as MRINE models of spike-LFP signals for every combination of these multimodal channel sets, and decoded the target behavior from inferred latent factors (see Appendix A.5.3 for details). In our analyses, we used 4 and 3 experimental sessions recorded on different days for the NHP grid reaching and center-out reaching datasets, respectively, and performed 5-fold cross-validation for each session. In our analyses, LFP and spike signals had different timescales: spikes were observed every 10 ms

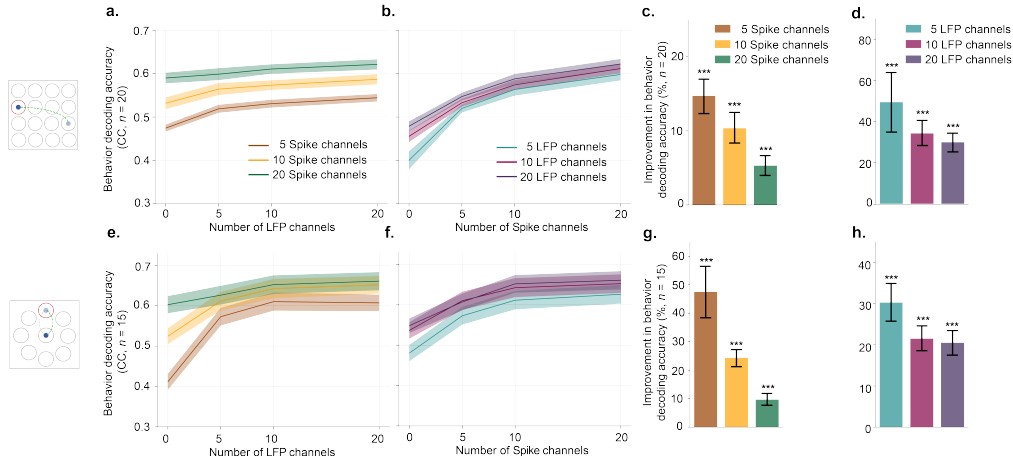

Figure 3: Behavior decoding accuracies for the NHP grid reaching dataset (*top row*) and NHP center-out reaching dataset (*bottom row*). Shaded areas and error bars represent SEM. **a.** Accuracies, when 5, 10, or 20 spike channels were the primary modality and an increasing number of LFP channels, were fused. **b.** Similar to **a** when LFP channels were the primary modality. **c.** Percentage improvements in decoding accuracy when 20 LFP channels were added to 5, 10, and 20 spike channels. Asterisks indicate significance of comparison (***: $p < 0.0005$, one-sided Wilcoxon signed-rank test). **d.** Similar to **c**, when 20 spike channels were added to 5, 10, and 20 LFP channels. **e–h**. Same as **a–d** but for NHP center-out reaching dataset.

and LFPs every 50 ms (abbreviated as 50 ms LFPs) [6, 28]. We found that when spiking signals were taken as the primary modality, fusing them with increasing numbers of LFP channels improved the behavior decoding accuracy for all different numbers of primary spike channels (Fig. 3a for NHP grid reaching dataset, $p < 0.009$, $n = 20$; Fig. 3e for NHP center-out reaching dataset, $p < 10^{-5}$, $n = 15$, one-sided Wilcoxon signed-rank test). Similar to simulation results, improvements in behavior decoding accuracy were higher in the low-information regimes, i.e., for 5 and 10 primary spike channels. For instance, adding 20 LFP channels to 5, 10, and 20 spike channels improved behavior decoding accuracy by $14.7\%$, $10.4\%$, and $5.3\%$, respectively, for the NHP grid reaching dataset (Fig. 3c), and $47.3\%$, $24.4\%$, and $9.8\%$ for the NHP center-out reaching dataset (Fig. 3g).

When LFP signals were considered as the primary modality, behavior decoding accuracies again improved significantly when spike channels were fused (Fig. 3b for grid reaching dataset, $p < 10^{-4}$, $n = 20$; Fig. 3f for center-out reaching dataset, $p < 10^{-5}$, $n = 15$, one-sided Wilcoxon signed-rank test). For example, adding 20 spike channels to 5, 10 and 20 LFP channels improved behavior decoding accuracy by $49.8\%$, $34\%$, and $29.6\%$ in the grid reaching dataset (Figs. 3d) and by $30.1\%$, $21.6\%$, and $20.6\%$ in the center-out reaching dataset (Fig. 3h). Overall, the bidirectional improvements—regardless of whether spikes or LFPs were the primary modality—for both datasets suggest that spike and LFP modalities may encode non-redundant information about behavior variables that can be fused nonlinearly with MRINE. Interestingly, adding spike channels to LFP

channels helped more than adding LFP channels to spike channels in the grid-reaching dataset, but the reverse held in the center-out reaching dataset. This suggests that while these modalities may have different amounts of behaviorally relevant information depending on the experiment (e.g., task or electrode types/locations), MRINE can combine their complementary strengths and be robust to such differences. Finally, we also tested MRINE when both modalities had the same timescale (10 ms) and found that MRINE again enabled consistent bidirectional improvements in behavior decoding performance for both datasets (Fig. 4).

## 4.3 MRINE improved behavior decoding compared with prior multimodal modeling methods

We compared MRINE with prior multimodal methods, namely the recent MSID in [28], mmPLRNN in [29], MMGPVAE in [31], and MVAE in [42] for both NHP datasets. We trained all models with 5, 10, or 20 channels of 10 ms spikes and 50 ms LFPs. Since mmPLRNN and MMGPVAE do not support training and inference with different timescale signals, we imputed the missing LFP timesteps by their global mean, i.e., zeros due to z-scoring, as is common practice. Further, mmPLRNN and MMGPVAE decodings were performed non-causally unlike MRINE and MSID since mmPLRNN and MMGPVAE do not support real-time recursive inference. For all numbers of primary channels (i.e., all information regimes), MRINE significantly outperformed all baselines in behavior decoding (Table 1, $p < 10^{-5}$, $n = 20$ for NHP grid reaching dataset; $p < 10^{-5}$, $n = 15$ for NHP center-out reaching dataset, one-sided Wilcoxon signed-rank test).

| Method | NHP grid reaching | | | NHP center-out reaching | | |
|---|---|---|---|---|---|---|
| | 5 Spike 5 LFP | 10 Spike 10 LFP | 20 Spike 20 LFP | 5 Spike 5 LFP | 10 Spike 10 LFP | 20 Spike 20 LFP |
| MVAE | $0.326 \pm 0.011$ | $0.386 \pm 0.009$ | $0.425 \pm 0.009$ | $0.392 \pm 0.017$ | $0.462 \pm 0.014$ | $0.544 \pm 0.018$ |
| MSID | $0.380 \pm 0.021$ | $0.440 \pm 0.015$ | $0.519 \pm 0.012$ | $0.442 \pm 0.022$ | $0.521 \pm 0.023$ | $0.561 \pm 0.020$ |
| mmPLRNN | $0.450 \pm 0.010$ | $0.477 \pm 0.013$ | $0.540 \pm 0.011$ | $0.468 \pm 0.022$ | $0.470 \pm 0.029$ | $0.538 \pm 0.032$ |
| MMGPVAE | $0.276 \pm 0.021$ | $0.438 \pm 0.019$ | $0.479 \pm 0.017$ | $0.495 \pm 0.020$ | $0.567 \pm 0.020$ | $0.601 \pm 0.021$ |
| MRINE | $\mathbf{0.487 \pm 0.007}$ | $\mathbf{0.555 \pm 0.011}$ | $\mathbf{0.611 \pm 0.012}$ | $\mathbf{0.547 \pm 0.020}$ | $\mathbf{0.624 \pm 0.022}$ | $\mathbf{0.649 \pm 0.021}$ |

Table 1: Behavior decoding accuracies for the NHP grid reaching and center-out reaching datasets with 5, 10, and 20 channels of 10 ms spikes and 50 ms LFP for MVAE, MSID, mmPLRNN, MMGPVAE, and MRINE. The best-performing method is in bold, the second best-performing method is underlined, $\pm$ represents SEM.

Next, using ablation studies on MRINE, we found that our multiscale encoder design was key in achieving higher performance than baselines. In particular, accounting for different timescales using the common technique of zero-imputation instead of using our multiscale encoder significantly degraded performance (Appendix A.6.4). Also, we performed the same baseline comparisons for the same timescale scenario (10 ms) and found that MRINE again outperforms all baseline methods for both datasets (Table 7). The degradation in performance due to having different timescales vs. having the same timescale was lower in MRINE than in MMGPVAE and mmPLRNN, again highlighting the importance of MRINE's design, especially when modalities have different timescales.

Finally, we trained single-scale models, MSID, MMGPVAE, and MRINE models on high-channel information regimes beyond 20 channels of spike and LFP signals, i.e., 30 and 60 channels. As shown in Table 8, MRINE again outperformed all methods, highlighting that MRINE's aggregation capabilities extend effectively to higher-dimensional recordings and that MRINE maintains performance advantages even as the information content increases. Please also see Appendix A.5.9 for trial-averaged latent factor visualizations, Appendix A.2 for more details on MVAE, MSID, mmPLRNN, and MMGPVAE benchmarks, and Table 9 for decoding accuracies using the $R^2$ metric instead of CC, showing that MRINE's improved performance is consistent across metrics.

## 4.4 MRINE's information aggregation was robust to missing samples

Next, we studied the robustness of all methods' inferences to randomly missing samples. Here, in addition to having different timescales, we used various sample-dropping probabilities to drop spike or LFP samples in both datasets. In both datasets, for models that were trained with 20 channels of 10 ms spikes and 50 ms LFP, we either fixed the dropping probability for LFP samples as 0.2 while varying that of spike samples (Fig. 5a,c) or vice versa (Fig. 5b,d) during inference. For the NHP grid reaching dataset, behavior decoding accuracies of MRINE remained robust, for example decreasing

by only 5.4% and 20.4% when 40% and 80% of spike samples were missing, respectively, in addition to 20% of LFP samples missing (Fig. 5a). Also, MRINE outperformed all baseline methods across all sample dropping regimes (Fig. 5a,b). Similarly for the NHP center-out reaching dataset, MRINE again outperformed all baseline methods across all sample dropping regimes (Figs. 5c,d).

In addition to behavior decoding, we also evaluated all methods' information aggregation capabilities in reconstructing spike and LFP signals in the same missing sample scenarios as for the previous case. As shown in Table 10, MRINE achieved competitive performance in reconstructing neural modalities as it achieved the best total average rank in terms of how well it reconstructed the neural modalities compared to baseline methods (see Appendix A.5.8 for details; averaged over the sample dropping probability regimes, modalities, and datasets in Fig. 6). This shows MRINE's capabilities beyond behavior decoding for neural reconstruction.

### 4.5 MRINE's performance improvements generalized to a distinct high-dimensional multimodal neural dataset with visual stimuli

To provide evidence for MRINE's generalization beyond neural datasets during motor tasks, we evaluated MRINE on a multimodal high-dimensional (i.e., 800-D) neural dataset that contains neuropixel spiking activity and calcium imaging data recorded from the visual cortex during visual stimuli presentation [67, 68] (see Appendix A.5.10 for details). This dataset contained neuropixel and calcium imaging data with different timescales (i.e., sampled at 120 and 30 Hz, respectively), which were subsequently smoothed with causal Gaussian kernels with standard deviations of 8 ms. In this analysis, we also included CEBRA [17] in our evaluations (see Appendix A.2.8 for CEBRA training details), as well as single-scale models trained on either modality. As the downstream task, we used the frame ID decoding task (0-900, 30Hz, 30s movie), similar to [17]. Since the downstream task was of slower frequency (30 Hz rather than 120 Hz), we performed average pooling on the latent factors inferred by MRINE for downsampling. As shown in Table 11, MRINE successfully aggregated information across neuropixel spiking activity and calcium imaging modalities, which had different timescales, and improved frame ID prediction performance over single-scale models. Furthermore, MRINE outperformed the CEBRA baseline, potentially due to its explicit dynamical backbone that can capture temporal dependencies in sequential time-series data, whereas CEBRA's convolutional architecture does not explicitly model state evolution.

## 5 Discussion

In this study, we presented MRINE that can dynamically, nonlinearly and in real-time aggregate information across multiple time-series modalities with different timescales or even with missing samples. To achieve this, we proposed a novel multiscale encoder design that first extracts modality-specific representations in real-time while accounting for their timescale differences and missing samples, and then performs nonlinear fusion in real-time to aggregate multimodal information. Through stochastic Lorenz attractor simulations and real NHP datasets, we show MRINE's ability to fuse information across modalities with different (or the same) timescales and with missing samples to achieve better downstream target decoding performance. We show that MRINE outperforms recent linear and nonlinear multimodal methods. Further, using these comparisons in addition to several ablation studies, we show the importance of MRINE's multiscale nonlinear encoder design and training objective outlined in Section 3.3 to enable accurate real-time fusion for multiscale data. A current limitation of MRINE is the assumption of time-invariant multiscale dynamics, which may not hold in non-stationary cases, and MRINE models may need to be intermittently retrained across days/sessions. Extending MRINE to track temporal variability, such as with switching dynamics or adaptive approaches, is an important future direction [13, 69–73]. Another promising direction is to incorporate automatic hyperparameter tuning methods, since MRINE introduces additional scaling terms for smoothness regularization penalties that we showed are important contributing factors to its improved downstream decoding performance. A further direction is to explore alternative training objectives for accurate cross-modal generation, which could allow MRINE and other baseline multimodal methods to reconstruct an entirely missing modality from the available one. Finally, incorporating external inputs in the MRINE model to disentangle input dynamics from intrinsic neural dynamics, as done in prior single-scale models [20, 21], is an interesting direction. Overall, MRINE can be especially important for applications where target variables are encoded across time-series modalities with different timescales and distribution, and/or where decoding needs to be done in real-time, for example, in brain-computer interfaces [74–76].

## Acknowledgements

This work was partly supported by the National Institutes of Health (NIH) grants RF1DA056402, R01MH123770, and R61MH135407, the NIH Director's New Innovator Award DP2-MH126378, and the National Science Foundation (NSF) CRCNS program award IIS-2113271. We sincerely thank all members of the Shanechi lab for helpful discussions.

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

# A Appendix

## A.1 Time-dropout

To improve the robustness of the MRINE against missing samples, we developed a regularization technique denoted as "*time-dropout*". Before training MRINE, based on the availability of the observations denoted by $\mathcal{T}$ and $\mathcal{T}'$ (see Section 3), we define mask vectors $\boldsymbol{m}^s, \boldsymbol{m}^y \in \mathbb{R}^T$ for $\boldsymbol{s}_t$ and $\boldsymbol{y}_t$ respectively:

$$m_t^y = \begin{cases} 1, & \text{if } \boldsymbol{y}_t \text{ is observed at } t \text{ (i.e., if } t \in \mathcal{T}' \\ 0, & \text{otherwise} \end{cases} \tag{13}$$

$$m_t^s = \begin{cases} 1, & \text{if } \boldsymbol{s}_t \text{ is observed at } t \text{ (i.e., if } t \in \mathcal{T} \\ 0, & \text{otherwise} \end{cases} \tag{14}$$

where subscript $t$ is the $t^{\text{th}}$ component of the vector. The general assumption is that $\mathcal{T}' \subseteq \mathcal{T}$, such that $\boldsymbol{s}_t$ is available for all time-steps where $\boldsymbol{y}_t$ is observed, motivated by the faster timescale of spikes compared with field potentials. However, this scenario may not always hold as recording devices can have independent failures leading to dropped samples at any time. To mimic the partially missing scenario where either modality can be missing, as well as the fully missing scenarios where both can be missing, we randomly replaced (dropped) elements of $\boldsymbol{m}^s$ and $\boldsymbol{m}^y$ by 0 at every training step, with the same dropout probabilities $\rho_t$ for both modalities. Masked time-steps (time-steps with 0 mask value) were not used either during the latent factor inference described in Section 3.2 or in the computation of loss terms in Eqs. 8 and 9 so that the inference and model learning were not distorted by missing samples. Instead, note that our inference procedure uses the learned model of dynamics to account for missing samples during both inference and learning. We note that we used the original masks (before applying *time-dropout*) while computing $\mathcal{L}_{sm}$ in Eq. 10, as we wanted to obtain smooth representations for all available observations. Note that *time-dropout* differs from masked training that is commonly used for training transformer-based networks. Masked training aims to predict masked samples from existing samples unlike our training objective (see Section 3.3 for details). Here, the goal of *time-dropout* is to increase the robustness to missing samples by artificially introducing partially and fully missing samples during model training, rather than being the training objective itself. See the ablation study on the effect of *time-dropout* in Appendix A.6.1 which shows that MRINE models trained with *time-dropout* had more robust behavior decoding performance and the effect of *time-dropout* was more prominent with more missing samples.

In addition to the *time-dropout*, we also applied regular dropout [77] in the encoder's input and output layers with probability $\rho_d$.

## A.2 Training details

### A.2.1 Training single-scale networks

For the case of single-scale networks, we use a special case of the architecture in Fig. 1a by replacing the multiscale encoder with an MLP encoder, and by just using a single-scale LDM with one modality's decoder network. In particular, for field potentials, we use a Gaussian decoder, and for the spikes we use a Poisson decoder to obtain the corresponding likelihood distribution parameters. Also, instead of the multiscale LDM of MRINE, this MLP is then followed by a single-scale LDM to perform inference both in real-time and non-causally. During learning of the single-scale networks, we dropped the terms related to the discarded modality from loss functions in Eqs. 8, 9 and 10.

### A.2.2 Setting the likelihood scaling parameter $\tau$

The log-likelihood values of modalities with different likelihood distributions may be of different scales, e.g., Poisson log-likelihood has a smaller scale than Gaussian log-likelihood in our case (and this can change arbitrarily by simply multiplying the Gaussian modality with any constant). To prevent the model from putting more weight on one modality vs. the other due to a higher log-likelihood scale while learning the dynamics, we scaled the log-likelihood of the modality with a smaller scale by a parameter $\tau$. To set this parameter, we first computed the time averages of each

modality over $\mathcal{T}$ and $\mathcal{T}'$ for $s_t$ and $y_{t'}$, respectively, as:

$$\boldsymbol{\lambda} = \frac{1}{|\mathcal{T}|} \sum_{t \in \mathcal{T}} s_t \tag{15}$$

$$\boldsymbol{\mu} = \frac{1}{|\mathcal{T}'|} \sum_{t' \in \mathcal{T}'} y_{t'} \tag{16}$$

where $\boldsymbol{\lambda} \in \mathbb{R}^{n_s}$ and $\boldsymbol{\mu} \in \mathbb{R}^{n_y}$. Then, we computed the corresponding log-likelihoods of $s_t$ and $y_{t'}$ by using these time averages as the means of their likelihood distributions (assuming unit variance for Gaussian distribution). Finally, we set $\tau$ as the ratio between the higher and smaller scale log-likelihoods:

$$\tau = \frac{\frac{1}{|\mathcal{T}'|} \sum_{t' \in \mathcal{T}'} \log(\mathcal{N}(y_{t'}; \boldsymbol{\mu}, \mathbf{1}))}{\frac{1}{|\mathcal{T}|} \sum_{t \in \mathcal{T}} \log(\mathrm{Poisson}(s_t; \boldsymbol{\lambda}))} \tag{17}$$

where $\mathbf{1} \in \mathbb{R}^{n_y}$ is the 1's vector denoting the unit variances for the Gaussian likelihood. The above allows us to balance the contribution of both modalities in our loss function during learning.

### A.2.3 Hyperparameters and codebase

Tables 2, 3, 4 and 5 provide the hyperparameters and network architectures used for training single-scale networks and MRINE on stochastic Lorenz simulations and analyses of the real NHP datasets. Our codebase is available at `https://github.com/ShanechiLab/mrine`.

For all models, we used a cyclical learning rate scheduler [78] starting with a minimum learning rate of 0.001, and reaching the maximum learning rate of 0.01 in 10 epochs. The maximum learning rate is exponentially decreased by a scale of 0.99. Across all experiments, batch size was set to 32, MLP weights were initialized by Xavier-normal initialization [79], and tanh function was used as the activation function of hidden layers. All models were trained on CPU servers (AMD Epyc 7513 and 7542, 2.90 GHz with 32 cores) with parallelization.

For scaling hyperparameters of smoothness regularization penalty, the hyperparameter search on each dataset is performed using a random inner-training and inner-validation split from the training set of the first fold on the first available session over a small grid. For scaling hyperparameters of the Poisson and Gaussian modalities' smoothness regularization penalties ($\gamma_s$ and $\gamma_y$), we used a grid of [0, 30, 50, 100, 250] and [0, 5, 10, 50], respectively. For initial experiments, we recommend using smaller scaling hyperparameters, such as 30 and 5 for Poisson and Gaussian modalities, respectively. For the scaling hyperparameter of the smoothness penalty on $x_t$ ($\gamma_x$), we used a grid of [0, 30, 50] after finding $\gamma_s$ and $\gamma_y$ (in order to control the depth of hyperparameter search). Similarly, we recommend using $\gamma_x = 30$ for initial experiments.

| Models | Hyperparameters | | | | | | | | | | | | | | | |
|---|---|---|---|---|---|---|---|---|---|---|---|---|---|---|---|---|
| | $\phi_s$ | $\phi_y$ | $\phi_m$ | $\theta_s$ | $\theta_y$ | $n_a$ | $n_x$ | $\mathcal{K}$ | $\rho_t$ | $\rho_d$ | GC | $\gamma_s$ | $\gamma_y$ | $\gamma_x$ | $\gamma_r$ | TE |
| SS-Poisson | 3,128 | - | - | 3,128 | - | 32 | 32 | 1,2,3,4 | 0.3 | 0.4 | 0.1 | 100 | - | 30 | 0.0001 | 200 |
| SS-Gaussian | - | 3,128 | - | - | 3,128 | 32 | 32 | 1,2,3,4 | 0.3 | 0.4 | 0.1 | - | 50 | 30 | 0.0001 | 200 |
| MRINE | 3,128 | 3,128 | 1,128 | 3,128 | 3,128 | 32 | 32 | 1,2,3,4 | 0.3 | 0.4 | 0.1 | 250 | 10 | 30 | 0.001 | 200 |

Table 2: Hyperparameters used for the stochastic Lorenz attractor simulations. SS denotes single-scale network. We represent the architecture's various MLP encoders and decoders with their parameter notations, and for each, provide the number of hidden layers and hidden units in order, separated by commas. Specifically. $\phi_s$ and $\phi_y$—i.e., MLP blocks through which $s_t$ and $y_{t'}$ are passed in Fig. 1b—represent the modality-specific encoder networks for Poisson and Gaussian modalities, respectively. $\phi_m$ is the fusion network (the last MLP block in Fig. 1b). Modality-specific decoder networks are $\theta_s$ and $\theta_y$ for Poisson and Gaussian modalities, respectively. $n_a$ and $n_x$ represent dimensions of $a_t$ and $x_t$, respectively. $\mathcal{K}$ is the set of future prediction horizons in Eq. 8. $\rho_t$ is the time dropout probability on the mask vectors, and $\rho_d$ denotes the dropout probability applied in the input and output layers of the encoder network (see Appendix A.1). GC represents the global gradient clipping norm on learnable parameters. $\gamma_s$, $\gamma_y$, and $\gamma_x$ are scaling parameters for smoothness regularization penalty in Eq. 10. $\gamma_r$ is the $L_2$ penalty on MLP weights of the encoder and decoder networks. TE denotes the number of training epochs.

| Models | Hyperparameters | | | | | | | | | | | | | | | |
|---|---|---|---|---|---|---|---|---|---|---|---|---|---|---|---|---|
| | $\phi_s$ | $\phi_y$ | $\phi_m$ | $\theta_s$ | $\theta_y$ | $n_a$ | $n_x$ | $\mathcal{K}$ | $\rho_t$ | $\rho_d$ | GC | $\gamma_s$ | $\gamma_y$ | $\gamma_x$ | $\gamma_r$ | TE |
| SS-Poisson | 3,128 | - | - | 3,128 | - | 64 | 64 | 1,2,3,4 | 0.3 | 0.1 | 0.1 | 100 | - | 30 | 0.0001 | 500 |
| SS-Gaussian | - | 3,128 | - | - | 3,128 | 64 | 64 | 1,2,3,4 | 0.3 | 0.1 | 0.1 | - | 10 | 30 | 0.0001 | 500 |
| MRINE | 3,128 | 3,128 | 1,128 | 3,128 | 3,128 | 64 | 64 | 1,2,3,4 | 0.3 | 0.1 | 0.1 | 250 | 10 | 30 | 0.001 | 500 |

Table 3: Hyperparameters used for the NHP grid reaching dataset analysis with same timescales for both modalities. Hyperparameter definitions are the same as in Table 2.

| Models | Hyperparameters | | | | | | | | | | | | | | | |
|---|---|---|---|---|---|---|---|---|---|---|---|---|---|---|---|---|
| | $\phi_s$ | $\phi_y$ | $\phi_m$ | $\theta_s$ | $\theta_y$ | $n_a$ | $n_x$ | $\mathcal{K}$ | $\rho_t$ | $\rho_d$ | GC | $\gamma_s$ | $\gamma_y$ | $\gamma_x$ | $\gamma_r$ | TE |
| SS-Poisson | 3,128 | - | - | 3,128 | - | 64 | 64 | 1,2,3,4 | 0.3 | 0.1 | 0.1 | 100 | - | 30 | 0.0001 | 500 |
| SS-Gaussian | - | 3,128 | - | - | 3,128 | 64 | 64 | 1,2,3,4 | 0.3 | 0.1 | 0.1 | - | 5 | 30 | 0.0001 | 500 |
| MRINE | 3,128 | 3,128 | 1,128 | 3,128 | 3,128 | 64 | 64 | 1,2,3,4 | 0.3 | 0.1 | 0.1 | 250 | 5 | 30 | 0.001 | 500 |

Table 4: Hyperparameters used for the NHP grid reaching dataset analysis with different timescales for the different modalities. Hyperparameter definitions are the same as in Table 2.

| Models | Hyperparameters | | | | | | | | | | | | | | | |
|---|---|---|---|---|---|---|---|---|---|---|---|---|---|---|---|---|
| | $\phi_s$ | $\phi_y$ | $\phi_m$ | $\theta_s$ | $\theta_y$ | $n_a$ | $n_x$ | $\mathcal{K}$ | $\rho_t$ | $\rho_d$ | GC | $\gamma_s$ | $\gamma_y$ | $\gamma_x$ | $\gamma_r$ | TE |
| SS-Poisson | 3,128 | - | - | 3,128 | - | 64 | 64 | 1,2,3,4 | 0.3 | 0.1 | 0.1 | 30 | - | 30 | 0.0001 | 200 |
| SS-Gaussian | - | 3,128 | - | - | 3,128 | 64 | 64 | 1,2,3,4 | 0.3 | 0.1 | 0.1 | - | 5 | 30 | 0.0001 | 200 |
| MRINE | 3,128 | 3,128 | 1,128 | 3,128 | 3,128 | 64 | 64 | 1,2,3,4 | 0.3 | 0.1 | 0.1 | 50 | 5 | 30 | 0.001 | 200 |

Table 5: Hyperparameters used for the NHP center-out reaching dataset analysis with the same and different timescales for the different modalities. Hyperparameter definitions are the same as in Table 2.

| Models | Hyperparameters | | | | | | | | | | | | | | | |
|---|---|---|---|---|---|---|---|---|---|---|---|---|---|---|---|---|
| | $\phi_s$ | $\phi_y$ | $\phi_m$ | $\theta_s$ | $\theta_y$ | $n_a$ | $n_x$ | $\mathcal{K}$ | $\rho_t$ | $\rho_d$ | GC | $\gamma_s$ | $\gamma_y$ | $\gamma_x$ | $\gamma_r$ | TE |
| SS-Gaussian | - | 3,128 | - | - | 3,128 | 64 | 64 | 1,2,3,4 | 0.3 | 0.1 | 0.1 | - | 0 | 0 | 0.001 | 500 |
| MRINE | 3,128 | 3,128 | 1,128 | 3,128 | 3,128 | 64 | 64 | 1,2,3,4 | 0.3 | 0.1 | 0.1 | - | 5 | 30 | 0.001 | 500 |

Table 6: Hyperparameters used for the visual stimuli dataset analysis. Hyperparameter definitions are the same as in Table 2. Note that both modalities were treated with a Gaussian observation model for MRINE, and single-scale models for both modalities were trained using the hyperparameter denoted for SS-Gaussian.

### A.2.4   MSID training

Multiscale subspace identification (MSID) is a recently proposed linear multiscale dynamical model of neural activity that assumes linear dynamics. MSID can handle different timescales and allow for real-time inference of latent factors [28]. We compared MRINE with MSID and showed that compared with the linear approach of MSID, the nonlinear information aggregation enabled by MRINE can improve downstream behavior decoding while still allowing for real-time recursive inference and for handling different timescales. To train MSID, we used the implementation provided by the authors and set the horizon hyperparameters with the values provided in their manuscript, i.e., $h_y = h_z = 10$. For the MSID results reported in this work, as recommended by the developers in their manuscript [28], we fitted MSID models with various latent dimensionalities consisting of [8, 16, 32, 64] and picked the one with the best behavior decoding accuracy found with inner-cross validation done on the training data.

### A.2.5   mmPLRNN training

Multimodal piecewise-linear recurrent neural networks (mmPLRNN) is a recent multimodal framework that assumes piecewise-linear latent dynamics coupled with modality-specific observation models [29]. As discussed in Section 2, mmPLRNN has shown great promise in reconstructing the underlying dynamical system but its inference network operates non-causally in time and assumes the same timescale between modalities, unlike MRINE. Therefore, to train mmPLRNN models with different timescale signals in Table 1, we performed global-mean imputation for the missing signals as done in common practice. Also, for mmPLRNN results reported in this work, the behavior decoding with mmPLRNN was performed non-causally. To train mmPLRNN, we used the variational inference training code provided by the authors in their manuscript. However, the default implementation of mmPLRNN only supports Gaussian and categorical distributed modalities. Thus, we implemented

the Poisson observation model by following Appendix C of [29]. Further, we replaced the default linear decoder networks with nonlinear MLP networks for a fair comparison and better performance.

Finally, we trained all mmPLRNN models for 100 epochs. We also performed hyperparameter searches for latent state dimension, learning rate, and number of neurons in encoder/decoder layers over grids of [16, 32, 64], [1e-4, 5e-4, 1e-3, 1e-2] and [32, 64, 128], respectively.

As shown in Tables 1 and 7, MRINE outperformed mmPLRNN in behavior decoding for both NHP datasets across all information regimes. Further, we observed that mmPLRNN experienced higher decline in behavior decoding performance when trained with different timescale signals compared to MRINE and MSID (which support training and inference with multiscale signals and missing samples), indicating the importance of multiscale modeling. We believe that the future-step-ahead prediction training objective along with new smoothness regularization terms and smoothed reconstruction are also important elements contributing to such performance gap (see Appendix A.6.2) whereas mmPLRNN is trained on optimizing evidence lower bound (ELBO), whose optimization can be challenging due to KL-divergence term [80–83].

### A.2.6 MVAE Training

Multimodal variational autoencoder (MVAE) is a variational autoencoder-based architecture proposed in [42] that can account for partially paired multimodal datasets by a mixture of experts posterior distribution factorization. However, the notion of partial observations in our work and MVAE are different. In MVAE, partial observations refer to having partially missing data tuples in each element of the batch, which would translate to having completely missing LFP or spike signals for a given trial/segment of multimodal neural activity. However, as we detailed in Section 3, we are interested in modeling multimodal neural activities with different sampling rates, i.e., partially missing time-steps rather than missing either of the signals completely. Further, as discussed in Section 2, the latent factor inference in MVAE is designed to encode each modality to a single factor, whereas MRINE is designed to infer latent factors for each time-step so that behavior decoding can be performed at each time-step. To account for all these differences, we trained MVAE models without a dynamic backbone by treating each time-step as a different data point that allowed us to train MVAE models with partially missing time-steps as done for MRINE. As shown in Table 1, MVAE showed the lowest overall performance among all methods which could be caused by lacking a dynamical backbone, unlike other methods.

### A.2.7 MMGPVAE training

Multimodal Gaussian process variational autoencoder (MMGPVAE) is another recent multimodal framework that utilizes Gaussian process to model latent distribution underlying multimodal observations [31]. Distinct from other approaches discussed in this work, MMGPVAE inference network extracts the frequency content of the latent factors followed by conversion to time domain representations, rather than direct estimation on the time domain. This approach allows MMGPVAE to prune high-frequency content in the latent factors that help in obtaining smooth representations. Similar to mmPLRNN, MMGPVAE does not allow training on modalities with different timescales, thus, for the comparisons provided in Table 1, we performed global-mean imputation for missing samples as common practice. Further, behavior decoding for MMGPVAE is performed non-causally for all MMGPVAE results reported in this work, as it does not support real-time latent factor inference. To train MMGPVAE, we used the authors' official implementation provided in their manuscript. To provide a fair comparison, we trained MMGPVAE models with 64-dimensional latent factors for each modality, where 32 dimensions of 64-dimensional latent factors were shared across modalities (i.e., MMGPVAE models in this work had 96 dimensional latent factors unlike other methods). All MMGPVAE models are trained for 100 epochs as behavior decoding performances reached their peak performance in around 100 epochs, then started degrading due to overfitting. The default encoder/decoder architecture for Gaussian modality in the MMGPVAE codebase was implemented for an image dataset, which resulted in poor performance in our dataset. Therefore, for the encoder/decoder architectures, we used the same architectures as MRINE. Further, we scaled the likelihood of Poisson modality with the same $\tau$ value as done for MRINE (see Section A.2.2). We also performed a hyperparameter search for the learning rate in a grid of [1e-3, 5e-4, 1e-4] since default values resulted in poor convergence. Despite MMGPVAE being the best competitor of MRINE as shown in Table 7, it also experiences a significant decline in its performance when trained

with different timescale signals (Table 1), which indicates the importance of multiscale modeling. Further, we believe that MRINE's training objective is another important factor contributing to its improved performance over MMGPVAE.

### A.2.8 CEBRA training

CEBRA is a contrastive learning–based model for neural activity with a convolutional encoder architecture [17]. To train CEBRA models for the NHP grid reaching task and the visual stimuli dataset, we followed the tutorial provided by the authors [2]. To train multimodal models of CEBRA, we followed the multisession CEBRA training protocol as done in the tutorial. For the visual stimuli dataset, we used the same hyperparameters as in the tutorial. For the NHP grid-reaching task, we used *offset-36* encoder models as they achieved better performance, and set the output dimensionality to 64 for fair comparison with other baseline methods.

### A.2.9 LFADS training

LFADS is a sequential autoencoder-based model of unimodal neural activity proposed by [3]. We used the authors' codebase [3] to train LFADS models. Also, we used the hyperparameters in the second row of Supplementary Table 1 in [3] with a factor dimension of 64 and used the default learning rate scheduler and early stopping criterion used in the codebase. We followed two different approaches to train LFADS models. First, we trained LFADS models on concatenated multimodal data, which corresponds to treating multimodal data as a single modality. Second, we implemented an extension of multimodal LFADS (denoted as LFADS-multimodal) by incorporating modality-specific decoders with appropriate Poisson and Gaussian observation models. Similar to MRINE, we applied scaling on the Poisson likelihood loss term (i.e., scaled by 3, which was roughly the automatically selected scale for MRINE) for balancing.

### A.3 Latent factor inference times

In addition to enabling real-time inference of latent factors through model design, it is also important to report per-timestep inference times, as these provide a direct measure of the practical efficiency of the approach. Thus, we computed per-timestep inference times of MRINE and baseline methods with an Intel i7-10700K processor, averaged over more than 25,000 timesteps. MRINE's latent inference time per timestep is $1.82 \pm 0.13$ ms, which is smaller than the timestep considered in this study (10 ms), and smaller than common brain-computer interface (BCI) timesteps (e.g., 50 ms in [54]), thus suggesting real-time BCI applicability. Also, the latent inference times per timestep are $1.83 \pm 0.12$ ms for MMGPVAE, $0.03 \pm 0.008$ ms for MSID, and $25.91 \pm 3.44$ ms for mmPLRNN. The multi-modal GP approach had a similar inference time compared to MRINE, and MSID achieved the fastest per-timestep latent factor inference due to its simple linear form.

### A.4 Stochastic Lorenz attractor simulations

### A.4.1 Dynamical system

The following set of dynamical equations defines the stochastic Lorenz attractor system:

$$
\begin{aligned}
dx_1 &= \sigma(x_2 - x_1)dt + q_1 \\
dx_2 &= (\rho x_1 - x_1 x_3 - x_2)dt + q_2 \\
dx_3 &= (x_1 x_2 - \beta x_3)dt + q_3
\end{aligned}
\tag{18}
$$

where $x_1$, $x_2$, and $x_3$ are the latent factors of the Lorenz attractor dynamics, $dt$ denotes the discretization time-step of the continuous system and $d$ is the change of variables in $dt$ time. We used $\sigma = 10$, $\rho = 28$, $\beta = \frac{8}{3}$ and $dt = 0.006$ as in [3]. $q_1$, $q_2$, and $q_3$ are zero-mean Gaussian dynamic noises with variances of 0.01. We generated 750 trials each containing 200 time-steps, and the initial condition of each trial was obtained by running the system for 500 burn-in steps starting from a random point. Then, we normalized trajectories to have zero mean and a maximum value of 1 across time for each latent dimension.

---

[2]https://cebra.ai/docs/demo_notebooks/Demo_Allen.html
[3]https://lfads.github.io/lfads-run-manager/

### A.4.2 Generating high dimensional observations

To generate the Gaussian-distributed modalities, we multiplied the normalized latent factors by a random matrix $C_y \in \mathbb{R}^{n_y \times 3}$ and added zero mean Gaussian noise with variance of 5 to generate noisy observations.

To generate the Poisson-distributed modalities, we first generated firing rates by multiplying the normalized trajectories by another random matrix $C_s \in \mathbb{R}^{n_s \times 3}$ and added a log baseline firing rate of $5$ spikes/sec with bin-size of 5 ms, followed by exponentiation. We then generated spiking activity by sampling from the Poisson process whose mean is the simulated firing rates.

### A.4.3 Computing the latent reconstruction accuracy

For MRINE and each single-scale network trained only on Poisson or Gaussian modalities, we obtained the smoothed single-scale or multiscale latent factors $x_{t|T}$, $t \in \{1, 2, \ldots, T\}$ for each trial in the training and test sets. To quantify how well the inferred latent factors can reconstruct the true latent factors, we fitted a linear regression model from the inferred latent factors of the training set to the corresponding true latent factors. Using the same linear regression model, we reconstructed the true latent factors from the inferred latent factors of the test set. Then, we computed the Pearson correlation coefficient (CC) between the true and reconstructed latent factors for each trial and latent dimension. The reported values are averaged over trials and latent dimensions.

## A.5 Real dataset analyses

### A.5.1 Nonhuman primate (NHP) grid reaching dataset

In this publicly available dataset [64, 65], a macaque monkey performed a 2D target-reaching task by controlling a cursor in a 2D virtual environment. All experiments were performed in accordance with the US National Research Council's Guide for the Care and Use of Laboratory Animals and were approved by the UCSF Institutional Animal Care and Use Committee. Monkey I was trained to perform continuous reaches to circular targets with a 5 mm visual radius randomly appearing on an 8-by-8 square or an 8-by-17 rectangular grid. The cursor was controlled by the monkey's fingertips, and the targets were acquired if the cursor stayed within a 7.5 mm-by-7.5 mm target acceptance zone for 450 ms. Even though there was no inter-trial interval between sequential reaches, there existed a 200 ms lockout interval after a target acquisition during which no target could be acquired. After the lockout interval, a new target was randomly drawn from the set of possible targets with replacement. Fingertip position was recorded with a six-axis electromagnetic position sensor (Polhemus Liberty, Colchester, VT) at 250 Hz and non-causally low-pass filtered to reject the sensor noise (4th order Butterworth, with 10 Hz cut-off frequency). The cursor position was computed by a linear transformation of the fingertip position, and we computed 2D cursor velocity using discrete differentiation of the 2D cursor position in the x and y directions. In our analysis, we used the 2D cursor velocity as the behavior variable to decode.

One 96-channel silicon microelectrode array (Blackrock Microsystems) was chronically implanted into the subject's right hemisphere primary motor cortex. Each array consisted of 96 electrodes, spaced at 400 $\mu$m and covering a 4mm-by-4mm area. We used multi-unit spiking activity obtained at a 10 ms timescale, and LFP signals were extracted from the raw neural signals by low-pass filtering with 300 Hz cut-off frequency, and downsampling to either 100 Hz (10 ms timescale) or 20 Hz (50 ms timescale). In our study, we picked the top spiking and LFP channels based on their individual behavior prediction accuracies and considered a maximum of 20 channels for each modality. As this dataset consists of continuous recordings without a clear trial structure, we created 1-second non-overlapping segments from continuous recordings to form trials so that we could utilize mini-batch gradient descent during model learning.

### A.5.2 NHP center-out reaching dataset

In this publicly available dataset [66], a macaque monkey performed a 2D center-out reaching task while grasping a two-link manipulandum. All experiments were performed with approval from the Institutional Animal Care and Use Committee of Northwestern University. Monkey C was trained to perform reaches from a center position to 2-cm square outer targets in an 8-target environment, where outer targets were spaced at 45-degree intervals around a 10-cm radius circle. Each trial of the task

started with the illumination of the center target where the monkey had to hold the manipulandum for a random hold time of 0.5-0.6 seconds. After, the center target disappeared and an outer target was randomly selected from the pool of possible 8 targets, which signaled the monkey to start the reach. To obtain the reward, the monkey had to reach the outer target within 1.5 seconds and hold the manipulandum at the outer target for a random time of 0.2-0.4 seconds. Then, the monkey returned back to the center target position and the next trial started. In our analysis, we used 2D manipulandum velocity as the behavior variable to decode.

One 96-channel silicon microelectrode array (Blackrock Microsystems) was chronically implanted into the subject's proximal arm area of primary motor (M1) and premotor (PMd) cortices contralateral to the arm used to perform the task. We used multi-unit spiking activity obtained at a 10 ms timescale. LFP signals in the original dataset were extracted from the raw neural signals by band-pass filtering between 0.5 and 500 Hz and sampled at 2 kHz. From these LFP signals, we computed LFP power signals with a window of size 256 ms (moved at 10 ms resolution) over 5 bands (0-4, 7-20, 70-115, 130-200 and 200-300 Hz), resulting in LFP power signals at 100 Hz. For the different timescale analyses, we downsampled LFP power signals to 20 Hz (50 ms timescale). The rest of the dataset generation details are the same as the previous dataset.

### A.5.3   Behavior decoding

In our analyses, we took 2D cursor or manipulandum velocity in the x and y directions as the behavior variables for downstream decoding. For all methods, after we inferred latent factors for both training and test sets, we fitted a linear regression model from inferred latent factors of the training set to the corresponding behavior variables. Then, we used the same linear regression model to decode the behavior variables from the inferred latent factors in the test set. We quantified the behavior decoding accuracy by computing the CC between the true and reconstructed behavior variables across time and averaging over behavior dimensions.

When MRINE was trained with spiking activity and LFP signals with timescales of 10 ms and 50 ms, respectively, behavior decoding was performed at the 10 ms timescale for comparisons between MRINE and single-scale networks trained with spike channels. To provide a fair comparison between MRINE and single-scale networks trained with 50 ms LFP signals, inferred latent factors of MRINE were downsampled to 50 ms from 10 ms, and behavior was decoded at every 50 ms in these comparisons with LFP. For all baseline comparisons, behavior decoding with MRINE is performed in real-time (i.e., multiscale latent factors are inferred via real-time/causal Kalman filtering) unless otherwise stated.

### A.5.4   Behavior decoding with same timescale signals

For both NHP grid reaching and NHP center-out reaching datasets, we also trained MRINE models with various combinations of 5, 10, and 20 channels of the same timescale spike and LFP signals. As shown in Fig. 4, for both datasets, MRINE again successfully improved behavior decoding accuracies as LFP channels are fused with primary spike channels (Figs. 4a,c for NHP grid reaching dataset and Figs. 4e,g for NHP center-out reaching dataset), and when spike channels are added to primary LFP channels (Figs. 4b,d for NHP grid reaching dataset and Figs. 4f,h for NHP center-out reaching dataset). These results provide further evidence of MRINE's information aggregation capabilities beyond its multiscale modeling.

Further, we performed the same baseline comparisons for all numbers of primary channels of the same timescale spike and LFP channels (similar to Table 1 for different timescale signals). For this analysis, we did not perform imputation for mmPLRNN and MMGPVAE models since both spike and LFP channels have the same timescale, i.e., 10 ms. Note that we did not include MVAE in this analysis since it is not a dynamical model and it had the lowest performance among all methods in Table 1.

As shown in Table 7, MRINE achieves the best behavior decoding performance both with its real-time and noncausal latent factor inference for the NHP grid reaching dataset across all information regimes ($p < 10^{-5}$, $n = 20$, one-sided Wilcoxon signed-rank test). For the NHP center-out reaching dataset, MRINE achieves the best behavior decoding performance among all methods with noncausal latent factor inference ($p < 0.006$, $n = 15$, one-sided Wilcoxon signed-rank test). When MRINE performs real-time (causal) latent factor inference, MMGPVAE outperforms MRINE across low (5 channels)

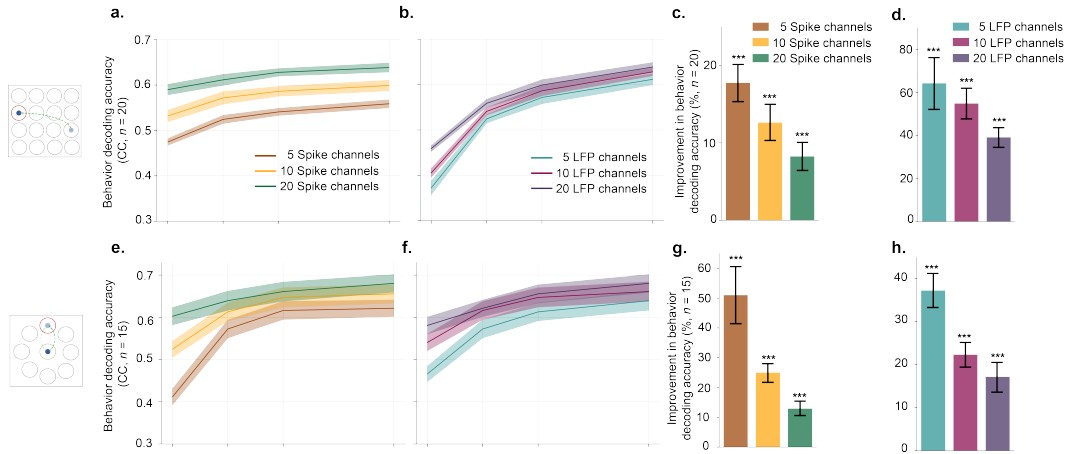

Figure 4: Behavior decoding accuracies for the NHP grid reaching dataset (*top row*) and NHP center-out reaching dataset (*bottom row*) with same timescale spike and LFP signals. Figure conventions are the same as in Fig. 3.

and high (20 channels) information regimes where MRINE is slightly better in medium (10 channels) information regime, however, improvements are not statistically significant ($p = 0.08$ for 5 channels and MMGPVAE > MRINE, $p = 0.12$ for 10 channels and MRINE > MMGPVAE, and $p = 0.11$ for 20 channels and MMGPVAE > MRINE, $n = 15$, one-sided Wilcoxon signed-rank test).

Beyond the nonlinear multiscale modeling capabilities of MRINE which are crucial for neuroscientific application purposes and are not achieved by recent prior nonlinear benchmark methods (see performance differences of mmPLRNN and MMGPVAE between Table 1 for different timescale results and Table 7 for same timescale results), these results suggest that MRINE achieves competitive performance through its encoder design and training objectives even when both modalities are recorded with the same timescale.

| Method | NHP grid reaching | | | NHP center-out reaching | | |
| --- | --- | --- | --- | --- | --- | --- |
| | 5 Spike 5 LFP | 10 Spike 10 LFP | 20 Spike 20 LFP | 5 Spike 5 LFP | 10 Spike 10 LFP | 20 Spike 20 LFP |
| MSID | $0.452 \pm 0.015$ | $0.483 \pm 0.015$ | $0.544 \pm 0.016$ | $0.467 \pm 0.019$ | $0.548 \pm 0.020$ | $0.596 \pm 0.022$ |
| mmPLRNN | $0.455 \pm 0.012$ | $0.478 \pm 0.011$ | $0.533 \pm 0.012$ | $0.530 \pm 0.022$ | $0.556 \pm 0.024$ | $0.591 \pm 0.027$ |
| MMGPVAE | $0.424 \pm 0.012$ | $0.511 \pm 0.014$ | $0.579 \pm 0.010$ | $\underline{0.558 \pm 0.022}$ | $\underline{0.624 \pm 0.022}$ | $\underline{0.670 \pm 0.021}$ |
| MRINE | $\underline{0.493 \pm 0.008}$ | $\underline{0.566 \pm 0.010}$ | $\underline{0.621 \pm 0.011}$ | $0.550 \pm 0.021$ | $0.628 \pm 0.022$ | $0.663 \pm 0.020$ |
| MRINE - noncausal | $\mathbf{0.524 \pm 0.009}$ | $\mathbf{0.586 \pm 0.011}$ | $\mathbf{0.639 \pm 0.010}$ | $\mathbf{0.572 \pm 0.022}$ | $\mathbf{0.647 \pm 0.023}$ | $\mathbf{0.681 \pm 0.021}$ |

Table 7: Behavior decoding accuracies for the NHP grid reaching and center-out reaching datasets with 5, 10, and 20 channels of same timescale (10 ms) spike and LFP signals for MSID, mmPLRNN, MMGPVAE, and MRINE (both with real-time and noncausal inference). The best-performing method is in bold, the second best-performing method is underlined, $\pm$ represents SEM.

### A.5.5 Behavior decoding with missing samples

In this analysis, we first trained all baseline methods with 20 spike and 20 LFP channels with different timescales (whose behavior decoding accuracies are shown in Table 1 when there were no missing samples). To test the robustness of each method to missing samples, we randomly dropped samples in time during inference with fixed sample dropping probabilities for both modalities. Then, we inferred latent factors at all time-steps using only the available observations after sample dropping and performed behavior decoding as described above. Note that even though time-series observations were missing in time, behavior variables were available for all time-steps and were decoded at all time-steps. As shown in Fig. 5, MRINE outperformed all baseline methods for both datasets as it can leverage learned single-scale dynamics to account for missing samples within each data modality.

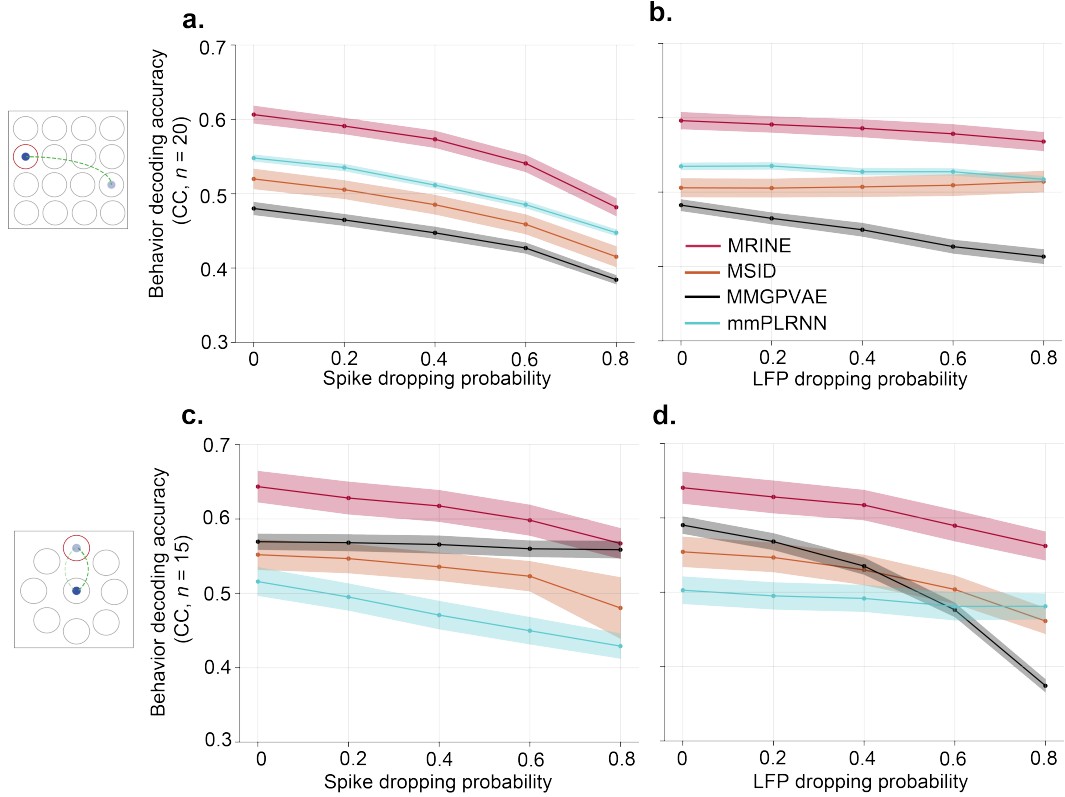

Figure 5: Behavior decoding accuracies all models for the NHP grid reaching dataset (*top row*) and NHP center-out reaching dataset (*bottom row*) when spike and LFP channels had missing samples and different timescales. **a.** Accuracies for models trained with 20 spike and 20 LFP channels. Sample dropping probability of LFPs was fixed at 0.2 while that of spikes was varied as shown on the x-axis. Lines represent mean and shaded areas represent SEM. **b.** Similar to **a** when sample dropping probability of spikes was fixed at 0.2 while that of LFPs was varied. **c, d**. Same as **a**, **b** but for NHP center-out reaching dataset.

### A.5.6 Behavior decoding with higher channel counts

In our analyses, we mainly focused on channel counts up to 20 due to several reasons. First, this low- to mid-information regime is quite important for testing multimodal aggregation capabilities, because the contribution from spikes and LFPs is more balanced and the added value of combining modalities can be assessed more directly. Second, implantable devices such as chronic BCIs can experience signal loss over time due to various factors [84, 85], such as scar tissue formation. Indeed, prior studies have shown that spiking activity from implanted electrodes can degrade faster than LFPs [86–89]. Thus, multimodal information aggregation is especially helpful for BCIs in these low- to mid-information regimes because such aggregation can allow patients to continue using their implanted BCI systems for extended periods. Specifically. multimodal aggregation can improve BCI accuracy and robustness even in the face of signal loss by combining spikes and LFPs, as also shown in prior studies [6, 28, 90]. Finally, some recording devices can have lower channel count. For example, recent wireless BCI systems have explored lower channel count designs to reduce bandwidth and power demands, such as a neural system-on-chip using 16 channels with on-chip feature extraction [91].

Nevertheless, to show that MRINE's information aggregation capabilities generalizes beyond low- to mid- information regimes, we also trained MRINE models on 30 spike and 30 LFP channels (30-30) as well as on 60 spike and 60 LFP channels (60-60) in the NHP grid reaching dataset, with the LFP and spikes having different timescales. For comparison, we also trained single-scale models, as well as MSID and MMGPVAE. In this scenario, we performed behavior decoding from predicted firing rates ($\lambda_t$) as this yielded better performance, unlike in the analyses with up to 20 channels, where

decoding directly from latent factors was more accurate. As shown in Table 8, MRINE outperformed the multimodal baselines and single-scale models across both 30-30 and 60-60 channel regimes, indicating that its information aggregation capabilities generalize to high-channel count (information) regimes.

| Method | 30 Spike 30 LFP | 60 Spike 60 LFP |
|---|---|---|
| SS LFP | $0.482 \pm 0.012$ | $0.456 \pm 0.013$ |
| SS Spike | $0.639 \pm 0.008$ | $0.637 \pm 0.006$ |
| MSID | $0.573 \pm 0.012$ | $0.597 \pm 0.010$ |
| MMGPVAE | $0.516 \pm 0.006$ | $0.568 \pm 0.006$ |
| MRINE | $\mathbf{0.676 \pm 0.011}$ | $\mathbf{0.693 \pm 0.010}$ |

Table 8: Behavior decoding accuracies for the NHP grid reaching dataset with 30-30 and 60-60 channels of 10 ms spikes and 50 ms LFP for single-scale models (SS), MSID, MMGPVAE, and MRINE. The best-performing method is in bold, the second best-performing method is underlined, ± represents SEM.

### A.5.7 Behavior decoding performance with $R^2$ metric

Throughout our analyses, we used CC as the main metric for comparing downstream behavior decoding performance across methods. In addition to this, we also computed the $R^2$ metric for the 20-channel regime results presented in Table 1 as an example to show that MRINE's superior performance is not an artifact of the reported metric. As shown in Table 9, MRINE again outperforms the baseline methods also when $R^2$ metric is used.

| Method | NHP grid reaching | | NHP center-out reaching | |
|---|---|---|---|---|
| | CC | $R^2$ | CC | $R^2$ |
| MVAE | $0.425 \pm 0.009$ | $0.190 \pm 0.009$ | $0.544 \pm 0.018$ | $0.298 \pm 0.022$ |
| MSID | $0.519 \pm 0.012$ | $0.273 \pm 0.013$ | $0.561 \pm 0.020$ | $0.343 \pm 0.030$ |
| mmPLRNN | $0.540 \pm 0.011$ | $0.302 \pm 0.013$ | $0.538 \pm 0.032$ | $0.294 \pm 0.036$ |
| MMGPVAE | $0.479 \pm 0.017$ | $0.334 \pm 0.012$ | $0.601 \pm 0.021$ | $0.351 \pm 0.029$ |
| MRINE | $\mathbf{0.611 \pm 0.012}$ | $\mathbf{0.375 \pm 0.013}$ | $\mathbf{0.649 \pm 0.021}$ | $\mathbf{0.435 \pm 0.030}$ |

Table 9: Behavior decoding CC and $R^2$ accuracies for the NHP grid reaching and center-out reaching datasets with 20 channels of 10 ms spikes and 50 ms LFP for MVAE, MSID, mmPLRNN, MMGPVAE, and MRINE. The best-performing method is in bold, the second best-performing method is underlined, ± represents SEM. CC results are the same as in Table 1.

### A.5.8 Neural reconstruction

To evaluate each method's information aggregation capabilities beyond behavior decoding, we computed reconstruction accuracies of both modalities under various sample dropping probabilities in addition to having different timescales. To do that, the modality of interest was randomly dropped with varying sample dropping probabilities when the other modality was dropped with a fixed 0.2 probability (Fig. 6). Therefore, the reconstructions of the missing modality were generated by leveraging the learned modality-specific and multiscale dynamics. For all methods, neural reconstructions were obtained with non-causal smoothing. For mmPLRNN and MMGPVAE, the missing timesteps for both spike and LFP signals were replaced by zeros, and the reconstruction metrics were computed between the true signals and model reconstructions of the missing timesteps. All models were trained with 20 spike and 20 LFP channels. For LFP signals modeled with Gaussian likelihood, we quantified the reconstruction accuracy by computing the CC between the reconstructed mean of the Gaussian likelihood distribution ($\boldsymbol{\mu}(\boldsymbol{a}_{t|T})$) and the true observations across time.

For spike signals modeled with Poisson likelihood, reconstruction accuracy was quantified using the area under the curve (AUC) of the receiver operating characteristic (ROC) measure [1, 24]. We constructed the ROC by using the reconstructed firing rates, i.e., $\boldsymbol{\lambda}(\boldsymbol{a}_{t|T})$, as the classification scores to determine whether a time-step contained a spike or not [92]. Both metrics were averaged over observation dimensions.

| Methods | | NHP grid reaching | | | NHP center-out reaching | | | Total Avg. |
|---|---|---|---|---|---|---|---|---|
| | | Spike | LFP | Avg. | Spike | LFP | Avg. | |
| | MSID | 2.6 | 3.0 | 2.8 | 2.8 | 3.2 | 3.0 | 2.9 |
| | mmPLRNN | 2.2 | 3.6 | 2.9 | 2.2 | 2.4 | 2.3 | 2.6 |
| | MMGPVAE | 3.8 | 2.4 | 3.1 | 4.0 | 1.8 | 2.9 | 3.0 |
| | **MRINE** | 1.4 | 1.0 | 1.2 | 1.0 | 2.6 | 1.8 | **1.5** |

Table 10: Neural reconstruction average ranks of each method for neural modalities and datasets. Average ranks for each neural modality and dataset (each cell) are computed by first ranking each method based on their reconstruction performances at a given sample dropping probability, and then averaging these ranks across the 5 sample dropping regimes in Fig. 6. Also, we compute the average rank across neural modalities within each dataset (denoted by Avg.) and across both datasets (denoted by Total Avg.). The best-performing method is in bold and the second best-performing method is underlined.

As shown in Table 10 and Fig. 6, MRINE achieved competitive performance compared to baselines. To quantify each method's success in reconstructing neural modalities, we computed each method's average rank in terms of how well it reconstructed the neural modalities compared with other methods (Total Avg. in Table 10). To compute these average ranks in Table 10, we first ranked each method based on its reconstruction performance for each given sample dropping probability, neural modality, and dataset. We then averaged these ranks across all sample-dropping probabilities, neural modalities, and datasets for each method to obtain the final average rank (Total Avg.). As shown in Fig. 6, there exists no method achieving the best performance across all sample dropping probabilities for both modalities and datasets. However, average ranks in Table 10 show that MRINE overall achieves the best average performance (Total Avg.) compared to baseline methods, indicating its competitive and robust neural reconstruction performance compared to baseline methods. Therefore, in addition to MRINE's superior performance in behavior decoding compared to baselines, these results show that MRINE is a valuable tool for neuroscientific studies that go beyond behavior decoding and study neural dynamics as well.

### A.5.9 Visualizations of latent dynamics

To compute trial-averaged 3D PCA visualizations, for each algorithm, we first computed 3D PCA projections of latent factors, split them based on trial start and end indices, interpolated them to a fixed length (due to variable-length trials), and then computed trial averages of PCA projections for each of 8 and 4 different reach directions for NHP grid reaching dataset and NHP center-out reaching dataset, respectively. As expected based on literature [93, 94], all models recovered rotational neural population dynamics (see Fig. 7 for NHP grid reaching dataset and Fig. 8 for NHP center-out reaching dataset). Among all these algorithms, MRINE had the clearest rotations while each method revealed noisier trajectories for the NHP center-out reaching dataset, potentially due to consisting of smaller numbers of trials whose latent factors are averaged to obtain trial-averaged trajectories.

### A.5.10 Visual stimuli dataset

To evaluate MRINE's generalizability beyond neural datasets during motor tasks, we trained MRINE models on a high-dimensional (i.e., 800-D) visual stimuli dataset that contained neuropixel spiking activity and calcium imaging data sampled at 120 Hz and 30 Hz, respectively [67, 68]. Before training the models, we applied Gaussian smoothing on modalities with a causal kernel with a standard deviation of 8 ms (and treated both modalities with Gaussian observation models). In addition to single-scale models trained on either modality, we also trained multisession CEBRA models using multimodal data. As the downstream task, we used the frame ID decoding task, as done in [17], where the goal is to predict the ID of the frame being shown to the subject, ranging from 1 to 900 (30 Hz and 30s movie). To align the latent factors of MRINE and CEBRA (extracted at 120 Hz following the faster modality) with the downstream target's frequency (30 Hz), we performed mean-pooling. For downstream decoding using each method's latent factors, we used k-nearest neighbor and Bayes classifiers, and picked the one that achieved better performance. In addition to single-scale models (rows 3 and 4), CEBRA and MRINE, we also trained downstream decoders directly on neural activity (rows 1 and 2). To quantify the prediction performance, we used the mean absolute error (MAE) between the true and predicted frame ID. As shown in Table 11, MRINE

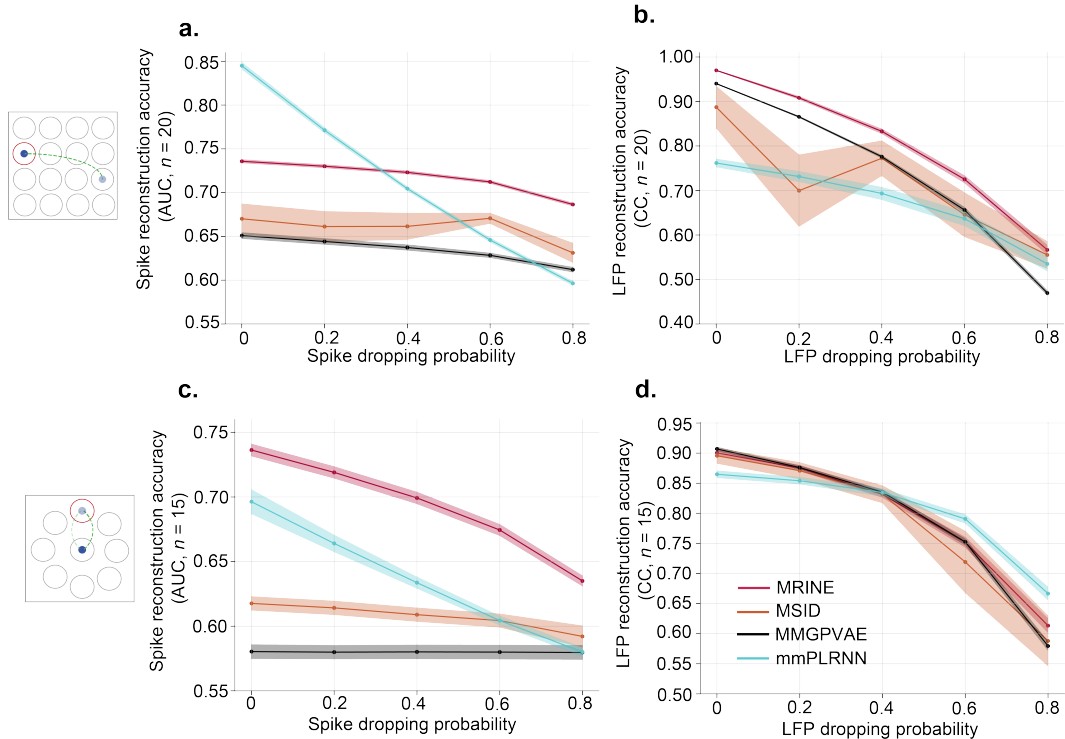

Figure 6: Neural reconstruction accuracies of 20 spike and 20 LFP channels for the NHP grid reaching dataset (*top row*) and NHP center-out reaching dataset (*bottom row*) when spike and LFP channels had different timescales. **a.** The reconstruction accuracies of the spike channels. Sample dropping probability of spikes was varied as shown on the x-axis while LFP channels were dropped with 0.2 probability. Lines represent mean and shaded areas represent SEM. **b.** Similar to **a** when sample dropping probability of LFPs was varied while spike channels were dropped with 0.2 probability. **c, d** Same as **a**, b but for NHP center-out reaching dataset.

successfully aggregated information across neuropixel spike and calcium imaging modalities of different timescales. Also, decoders trained on MRINE's latent factors improved the frame ID prediction performance over decoders trained on single-scale models' latent factors and directly on neural modalities. In addition, MRINE outperformed the CEBRA baseline, potentially due to its explicit dynamical modeling.

| Method | MAE |
|---|---|
| Calcium imaging | 151.60 |
| Neuropixel spike | 67.76 |
| SS Calcium imaging | 22.66 |
| SS Neuropixel spike | 10.94 |
| CEBRA | 9.31 |
| **MRINE** | **5.09** |

Table 11: Frame ID prediction performance for the visual stimuli dataset. SS denotes single-scale model. The best performing method is in bold, the second best performing method is underlined.

## A.6 Ablation Studies

### A.6.1 Effect of Time-Dropout

To test the effectiveness of *time-dropout*, for the NHP grid reaching dataset, we performed an ablation study with the same setting used to generate Figs. 5a,b (see Section 4.4) but we disabled *time-dropout* ($\rho_t = 0$). The remaining hyperparameters were as in Table 4. As shown in Fig. 9a, without *time-*

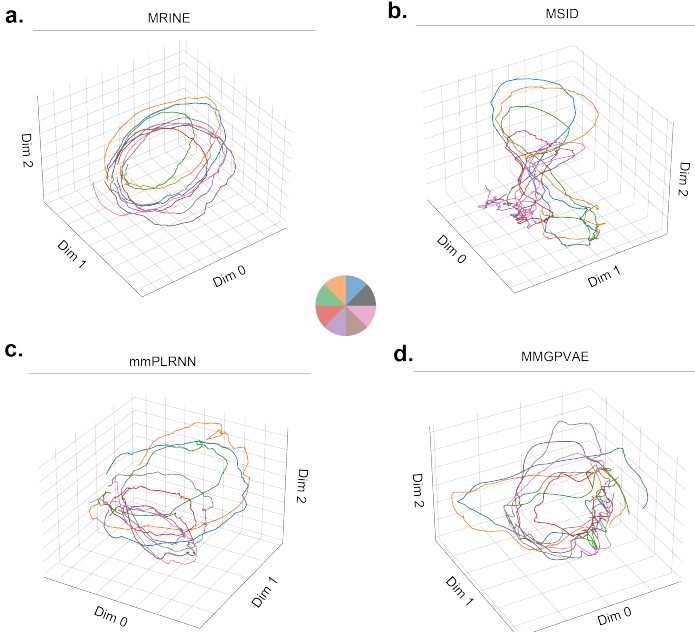

Figure 7: 3D PCA visualizations of trial-averaged latent factors inferred for NHP grid reaching dataset by **a)** MRINE, **b)** MSID, **c)** mmPLRNN, and **d)** MMGPVAE.

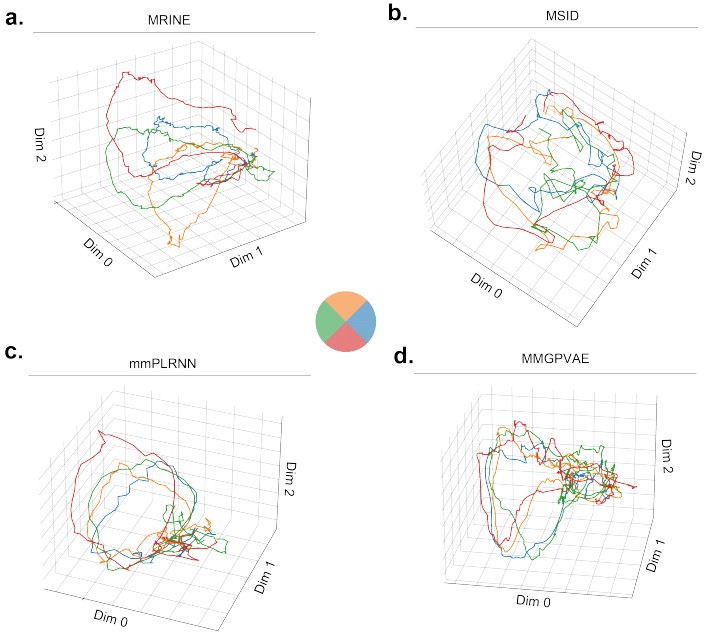

Figure 8: 3D PCA visualizations of trial-averaged latent factors inferred for NHP center-out reaching dataset by **a)** MRINE, **b)** MSID, **c)** mmPLRNN, and **d)** MMGPVAE.

*dropout*, the behavior decoding accuracies of MRINE decreased by 7.6% and 31.4% when 40% and 80% of spike samples were missing (in addition to 20% of LFP samples missing), whereas MRINE models trained with *time-dropout* experienced smaller performance drops of 5.4% and 20.4% in the same missing samples settings (see Figs. 5a,b). Similarly, MRINE models trained with *time-dropout* were more robust to missing LFP samples (Fig. 9b vs. Fig. 5b) but the performance drops were smaller due to spiking activity being the dominant modality for behavior decoding in this dataset. As

expected, the effect of *time-dropout* was more prominent in the high sample dropping probability regimes (i.e., more missing samples).

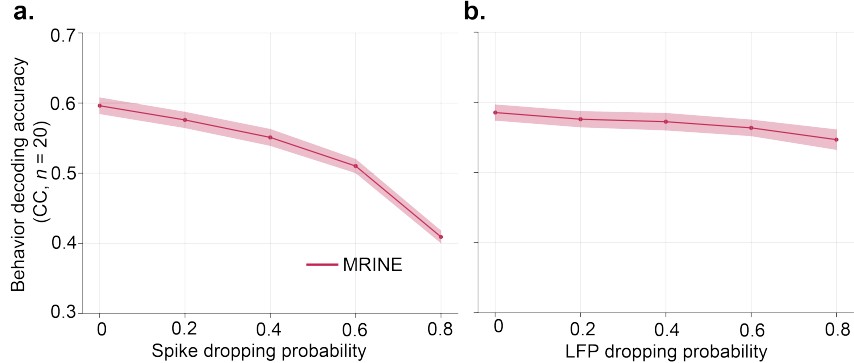

Figure 9: Behavior decoding accuracies in the NHP grid reaching dataset when *time-dropout* was disabled ($\rho_t = 0$), and spike and LFP channels had both missing samples and different timescales. Figure conventions are the same as in Fig. 5.

### A.6.2  Effect of loss terms in the behavior decoding performance

To gain intuition on the improved performance with MRINE, for the NHP grid reaching dataset, we performed an ablation study on the effect of smoothness regularization terms in Eq. 10 and smoothed reconstruction term in Eq. 9 on behavior decoding performance. To achieve that, we trained MRINE models with 20 channels of 10 ms spike and 20 channels of 10 ms LFP signals by removing Eq. 9 and individual terms in Eq. 10 from the training objective in Eq. 12. Then, we performed behavior decoding with these MRINE models as described in Section A.5.3.

As shown in Table 12, both smoothness regularization terms in Eq. 10 and smoothed reconstruction term in Eq. 9 are important factors contributing to improved behavior decoding performance as MRINE model trained without Eq. 9 and 10 (row 2) achieve worse performance than that of baseline methods in Table 7. We observed that applying smoothness regularization on $x_t$ is an important contributing factor to improved performance (row 3 vs row 5) as well as smoothing reconstruction term (row 2 vs row 3). Even though smoothness regularizations on $s_t$ and $y_{t'}$ may seem marginal when comparing results in rows 3 and 4, they play a crucial role when combined with smoothness regularization on $x_t$ (comparing row 1 and row 5).

| Model | Behavior Decoding (CC) |
|---|---|
| MRINE | **0.621 ± 0.010** |
| MRINE w/o Eq. 9 and Eq. 10 | 0.524 ± 0.013 |
| MRINE w/o Eq. 10 | 0.565 ± 0.012 |
| MRINE w/o $x_t$ in Eq. 10 | 0.566 ± 0.016 |
| MRINE w/o $s_t$ and $y_{t'}$ in Eq. 10 | 0.598 ± 0.012 |

Table 12: Behavior decoding accuracies for the NHP grid reaching dataset with 20 channels of 10 ms spike and 20 channels of 10 ms LFP signals for MRINE models trained without (w/o) loss terms denoted in the first column. The best-performing method is in bold, ± represents SEM.

### A.6.3 Effect of using different observation models

To better understand MRINE's performance compared to baseline methods, we trained MRINE models using Gaussian observation models for both modalities on the NHP grid reaching dataset. This approach allowed us to evaluate whether the use of distinct observation models for spike and LFP modalities or multiscale modeling is a primary source of MRINE's improvements. As shown in Table 13, modeling each modality with an appropriate observation model, i.e., Poisson for spikes and Gaussian for LFPs, significantly improves the performance as MRINE outperformed its variant where both modalities are modeled with the same (Gaussian) observation model ($p < 0.0003, n = 20$, one-sided Wilcoxon signed-rank test). Nonetheless, MRINE with the same observation model still outperformed the baseline methods shown in Table 1, indicating that MRINE's improved performance is mainly caused by its multiscale modeling and other elements in the training objective (see Appendix A.6.2).

Using the same observation model for both modalities can also enable a direct comparison to unimodal models, since the two modalities can simply be concatenated at the input level under a shared Gaussian observation model. To test this, we trained LFADS models using concatenated same timescale (i.e., 10 ms) spike and LFP signals of the NHP grid reaching dataset (see Appendix A.2.9 for details), as well as CEBRA models (see Appendix A.2.8 for details). In this scenario, MRINE achieved a downstream decoding CC of $0.621 \pm 0.011$, outperforming both LFADS ($0.549 \pm 0.012$), LFADS-multimodal ($0.547 \pm 0.011$), and CEBRA ($0.433 \pm 0.002$) models. These results further suggest that MRINE's superior performance stems from its multiscale architecture and training objective, rather than from simple input concatenation or observation model choices.

| Model | Behavior Decoding (CC) |
|---|---|
| MRINE | $0.611 \pm 0.012$ |
| MRINE w/ Same Observation Model | $0.604 \pm 0.011$ |

Table 13: Behavior decoding accuracies for the NHP grid reaching dataset with 20 channels of 10 ms spike and 20 channels of 50 ms LFP signals for MRINE trained with same and different observation models. $\pm$ represents SEM.

### A.6.4 Effect of multiscale encoder design

As discussed in Section 3.2, accounting for different sampling rates for neural signals is an important consideration for MRINE's encoder design shown in Fig. 1b. To achieve that, we learn modality-specific LDMs in MRINE's encoder that can leverage within-modality state dynamics to account for missing samples whether due to timescale differences or missed measurements. Therefore, MRINE can perform inference without relying on augmentations to impute missing samples, such as zero-imputation as done in common practice [53, 57] that can yield suboptimal performance [51, 52]. Such suboptimal performance regimes can include either degraded performance due to misinformation presented by imputed signals or discarding the imputed signal almost completely especially in the high imputation regimes due to attempting to reconstruct trivial imputations and focusing on only one neural modality. As shown in Tables 1 and 7, MRINE's performance degraded less than those of baseline methods when trained on different timescale signals due to these design choices.

To further investigate this, for the NHP grid reaching dataset, we trained MRINE models in a similar manner with baseline methods that do not account for training and inference on different timescale signals (i.e, mmPLRNN and MMGPVAE) where the missing LFP signals due to timescale difference are imputed by their global mean, i.e., zeros due to z-scoring. In this setting, missing LFP timesteps are discarded (masked) in the training objective (as done for mmPLRNN and MMGPVAE) but those timesteps were not treated as missing samples during latent factor inference both during training and inference. In other words, we let MRINE process the misinformation presented by zero imputation similar to mmPLRNN and MMGPVAE due to the recurrent nature of latent factor inference (even if their reconstructions/predictions are masked in the training objective).

In addition, we trained other versions of MRINE, mmPLRNN and MMGPVAE where imputed (missing) timesteps were not discarded in their training objectives (denoted by w/o Loss Masking in Table 14).

| Model | Behavior Decoding (CC) |
|-------|------------------------|
| MRINE | **0.611 ± 0.012** |
| MRINE w/ Zero Imputation | 0.581 ± 0.014 |
| MRINE w/ Zero Imputation and w/o Loss Masking | 0.523 ± 0.013 |
| mmPLRNN | 0.540 ± 0.011 |
| mmPLRNN w/o Loss Masking | 0.498 ± 0.009 |
| MMGPVAE | 0.479 ± 0.017 |
| MMGPVAE w/o Loss Masking | 0.500 ± 0.016 |

Table 14: Behavior decoding accuracies for the NHP grid reaching dataset with 20 channels of 10 ms spike and 20 channels of 10 ms zero-imputed LFP signals for MRINE, mmPLRNN, and MMGPVAE where zero-imputed LFP time-steps are either included or masked in the training objective. The best-performing method is in bold, ± represents SEM. MRINE, mmPLRNN, and MMGPVAE performances are taken from Table 1 for convenience.

As shown in Table 14, the performance of MRINE models trained with zero-imputed LFP signals degraded compared to MRINE models trained with 50 ms LFP signals (row 1 vs rows 2 and 3). As expected, masking zero-imputed LFP time-steps in the loss function improved behavior decoding performance compared to the scenario where they are included in the loss function (row 2 vs row 3). However, removing zero-imputed LFP time-steps from the training objective still results in degraded performance for MRINE, showing the importance of multiscale encoder design (row 1 vs row 2).

Even though the performance of mmPLRNN improved significantly when zero-imputed LFP time-steps were masked in the training objective (row 4 vs row 5), allowing it to achieve performance comparable to that in Table 1, unlike mmPLRNN, MMGPVAE achieved higher performance when imputed LFP signals were not masked in its training objective (row 7 vs. row 6). As discussed earlier, due to operating in a high imputation regime (i.e., 4 out of every 5 LFP signals are imputed), it is possible that including reconstruction of trivial imputed LFP signals may have led MMGPVAE to deprioritize LFP signals during latent factor inference, instead, focusing primarily on spike signals. This behavior suggests that fusing LFP signals with spiking signals can degrade MMGPVAE's performance, as including trivial LFP reconstructions in MMGPVAE's training objective results in higher performance. Therefore, a carefully designed encoder is essential to process both modalities effectively and maximize downstream performance.

Overall, MRINE significantly outperformed all models when they were trained with zero-imputed different timescale signals, including its own variants ($p < 0.02, n = 20$, one-sided Wilcoxon signed-rank test). These results show the importance of encoder design when modeling modalities with different timescales.

