# OpenReview forum: "Dynamical modeling of nonlinear latent factors in multiscale neural activity with real-time inference"
_NeurIPS.cc/2025/Conference — NeurIPS 2025 poster_

### Official Review · Reviewer_aE6q · 2025-06-19

**Clarity:** 3
**Significance:** 3
**Originality:** 3
**Rating:** 4
**Confidence:** 5

**Summary:**

In this paper, the authors introduces Multiscale Real-time Inference of Nonlinear Embeddings (MRINE) that is a novel learning framework designed for real-time, recursive decoding by dynamically and nonlinearly aggregating information from multiple neural time-series modalities.

It specifically addresses challenges posed by multiscale data, handling modalities with different timescales (e.g., fast spiking activity and slower LFP activity), varying probabilistic distributions (e.g., Poisson for spikes and Gaussian for LFP), and missing samples.

Key to MRINE is its novel multiscale encoder design, which first learns within-modality dynamics using Linear Dynamical Models (LDMs) to account for timescale differences and predict missing data, then performs nonlinear information fusion via a fusion network. The training objective incorporates a multi-horizon k-step-ahead prediction loss, smoothed reconstruction, and smoothness regularization on latent dynamics to encourage learning meaningful representations and prevent overfitting. It also introduces a time-dropout technique to further enhance robustness to missing samples during training.

Empirically, MRINE has demonstrated its ability to improve real-time target decoding in both stochastic Lorenz attractor simulations and two independent nonhuman primate (NHP) brain datasets. It outperforms various linear and nonlinear multimodal benchmarks, showing particular strength when modalities have different timescales or missing samples, and also exhibits robustness to randomly missing samples during inference.

**Questions:**

- What is the computational complexity (in Big O notation) per time step, and what is the actual computation time required per time step in practice?
- How effective is the model in terms of decoding performance when applied to reconstructed observations?

**Ethical Concerns:**

["NO or VERY MINOR ethics concerns only"]

**Final Justification:**

The authors have presented a strong and impactful contribution with MRINE, addressing critical challenges in decoding multiscale neural activity in real-time. The rebuttal effectively addressed most of the concerns raised, i.e.,
  Real-time inference times: Authors provided concrete per-timestep inference times, confirming MRINE's practical applicability for Brain-Computer Interfaces (BCIs).
 Evaluation methodology: The justification for using linear regression for latent factor decoding was well-explained, aligning with standard practices for assessing representational quality.
  Robustness to missing data: The added results on decoding from reconstructed inputs further highlight MRINE's versatility and strong handling of missing samples.
  Time-dropout distinction: Clarification on time-dropout as a regularization technique was helpful.
While acknowledged as a common limitation and future work, the assumption of time-invariant dynamics persists, which limits its direct applicability to highly non-stationary neural signals.
  Multiscale constraint assumption that the discrete modality evolves faster is a current design constraint, though the authors state the model can be extended.

**Limitations:**

Yes.

**Paper Formatting Concerns:**

None.

**Quality:**

3

**Strengths And Weaknesses:**

Strengths
*   The MRINE framework is designed for real-time recursive decoding from multiscale neural modalities. Its architecture, featuring a multiscale encoder with modality-specific Linear Dynamical Models (LDMs) and Kalman filtering, enables nonlinear information aggregation and intrinsically accounts for missing data, avoiding suboptimal zero-padding.
*    MRINE consistently outperforms various linear and nonlinear multimodal benchmarks (MSID, mmPLRNN, MMGPVAE, MVAE) in both stochastic Lorenz attractor simulations and nonhuman primate (NHP) brain datasets. The model demonstrates strong robustness against randomly missing spike or LFP samples during inference, outperforming baselines. It also achieves competitive performance in neural reconstruction.
*   Its real-time decoding capabilities and improved accuracy from multimodal fusion show potential for advanced brain-computer interfaces.

Weaknesses
* The model is based on the assumption of stationary linear dynamical systems, which may limit its capacity to effectively capture the complexity of neural dynamics.
* Multiscale constraint: The framework requires that one time index be a subset of the other, which may restrict modeling flexibility.
* Real-time performance: The paper lacks explicit discussion of computational efficiency or scalability. The term "real-time" appears to refer to the causal (filtering) nature of the inference procedure. To substantiate claims of real-time capability, the authors should demonstrate that the computational complexity per time step is constant with respect to the number of samples, and that the actual computation time per step does not exceed the duration of the time bin(s).
* Evaluation methodology: The paper does not clearly specify whether and how k-step prediction was applied to the non-human primate (NHP) datasets. As a result, it is uncertain whether the proposed approach is effective at long-term forecasting or primarily suited for short-term tracking.
* Comparison of latent factor decoding: The compared methods utilize different encoder and decoder architectures, typically involving nonlinear mappings between latent representations and observations. Therefore, evaluating performance based on linear decodability or Pearson correlation of latent factors may not provide a fair or meaningful comparison.
* Related work: A technique similar to time-dropout has previously been explored in https://openreview.net/forum?id=Ln8ogihZ2S.

Typos
- 727:  MMGVAE => MMGPVAE
- 857: Wilcoxon-signed => Wilcoxon signed-
- 920: most clear => clearest
- 981: missing sample => missing samples

---

> ### Author Rebuttal · Authors · 2025-07-30
>
> We thank the reviewer for their careful investigation of our work. Below, we address their comments.
>
> > Stationarity of neural signals
>
> We agree with the reviewer that accounting for non-stationarity is important for real-world decoding and discuss this as a limitation in the manuscript. Like many recent models—including MMGPVAE, mmPLRNN, LFADS, and CEBRA—MRINE assumes stationary encoding. Extending MRINE to handle distributional shifts or task-dependent changes is a valuable direction for future work, particularly for long-term or real-time applications. Indeed, developing calibration or alignment methods over time to address time variations is a complementary yet distinct direction of work (Farshchian et al., 2018; Karpowicz et al., 2025).
>
> > Multiscale constraint
>
> We indeed made an assumption on the multiscale nature of the signals by assuming that $T’$ is a subset of $T$, indicating that the discrete modality evolves faster, in line with the prior work. However, our model formulation can be extended to scenarios where $T’$ is not a subset of $T$. In those scenarios, we would represent both signals in the greatest-common-divisor timescale with appropriate masks to indicate the availability of the signal samples. For instance, if spike and LFP signals were sampled at 20 and 50 ms, we could represent each signal at 10 ms resolution during modeling, unmasking the spikes every other timestep and LFPs every 5 steps.
>
> > Real-time performance
>
> We thank the reviewer for their comment. We now report the per-timestep inference times for all methods to give an insight into the computational complexities. MRINE’s latent inference time per timestep is $1.82 \pm 0.13$ ms with Intel i7-10700K processor, which is smaller than the timestep considered in this study (10 ms), and smaller than common brain-computer interfaces (BCI) timesteps (e.g., 50 ms in Stavisky et al., 2015), thus suggesting real-time BCI applicability. Also, the latent inference times per timestep are $1.83 \pm 0.12$ ms for MMGPVAE, $0.03 \pm 0.008$ ms for MSID, and $25.91 \pm 3.44$ ms for mmPLRNN. Overall, we can say that the time complexity of MRINE’s inference is $O(T)$, where $T$ is the the number of timesteps, due to MRINE’s recursive nature. Also, the complexity per time-step is constant due to MRINE’s recursive nature.
>
> > Evaluation methodology
>
> We thank the reviewer for pointing this out. As detailed in Section 3.2 and L188-211, we perform k-step-ahead prediction of neural signals by first forward propagating the filtered multiscale latent factors ($x_{t|t}$) $k$-steps into the future through $x_{t+k|t} = A^k x_{t|t}$, and then obtain $k$-step-ahead predicted multisale embedding factors via $a_{t+k|t} = Cx_{t+k|t}$. These $k$-step-ahead predicted multiscale embedding factors are then passed through modality-specific decoder networks to obtain $k$-step-ahead prediction of input neural signals.
>
> That said, MRINE is primarily designed to decode target signals more accurately through multimodal information aggregation from modalities with different temporal resolutions, with real-time inference support. Thus, we mainly focused on downstream target decoding performance rather than short-term or future-forecasting of neural signals. This choice also influenced our design choices. For instance, we preferred linear dynamics instead of nonlinear dynamics or time-varying dynamics. Through this design choice, MRINE can allow for handling missing samples through its internal dynamics and can leverage both real-time Kalman filtering and noncausal Kalman smoothing without requiring to train separate models for different inference modes, whereas prior models support only one of the two.
>
> > Comparison of latent factor decoding
>
> We chose linear regression decoders to ensure a fair and standardized comparison of latent factor quality across models. This approach is widely used in both machine learning and neuroscience—for example, foundation models are often evaluated via linear probes, and in neuroscience, the Neural Latents Benchmark uses linear decoders to assess representational quality. For the specific example of Lorenz simulations, prior work also used similar linear metrics or alignment methods to quantify the alignment between inferred latent factors and true latent factors (attractors) through simple linear regressors or rotations (Sussillo et al., 2016; Gilpin, 2020).
>
> > Related work on time-dropout
>
> We thank the reviewer for bringing this excellent work to our attention that we will cite and discuss appropriately in the revised manuscript. While both approaches address missing data, we believe MRINE's use of time-dropout is conceptually distinct. Specifically, we introduce artificial sample dropping during training to improve robustness to missing inputs—analogous to dropout regularization. In contrast, Dowling et al. propose a method that explicitly models missing data during both training and inference, similar in spirit to Kalman filtering (i.e., setting Kalman gain to zero vs. $\alpha_t$ = 0). We will cite and discuss this work appropriately in the revised manuscript.
>
> > Decoding from reconstructed inputs
>
> We thank the reviewer for their question. In our main analyses, we decoded behavior from the latent factors, following common practice. As requested, we now also evaluated decoding performance from the reconstructed observations for the NHP grid-reaching task (20 spike + 20 LFP channels). Using reconstructed spiking rates, MRINE achieved a decoding CC of $0.621 \pm 0.011$, matching the performance from latent factors. As expected, decoding from reconstructed LFP signals yielded a lower CC of $0.490 \pm 0.012$, likely due to its lower behaviorally relevant information and signal to noise ratio.
>
> > Typos
>
> We sincerely thank the reviewer for bringing these typos to our attention. We will make sure to correct them.
>
> References:
>
> - Stavisky, S. D., Kao, J. C., Nuyujukian, P., Ryu, S. I., & Shenoy, K. V. (2015). A high performing brain–machine interface driven by low-frequency local field potentials alone and together with spikes. Journal of Neural Engineering.
> - Sussillo, D., Jozefowicz, R., Abbott, L. F., & Pandarinath, C. (2016). LFADS - Latent Factor Analysis via Dynamical Systems. arXiv.
> - Gilpin, W. (2020). Deep reconstruction of strange attractors from time series. Advances in Neural Information Processing Systems.

---

> > ### Comment · Reviewer_aE6q · 2025-08-03
> >
> > I thank the authors for addressing my questions and concerns in their rebuttal. I will maintain my score.

---

### Official Review · Reviewer_BeqV · 2025-06-28

**Clarity:** 2
**Significance:** 2
**Originality:** 2
**Rating:** 4
**Confidence:** 4

**Summary:**

The paper introduces MRINE, a method for real-time decoding of behavior from multimodal neural data with different timescales. By learning within-modality dynamics and fusing information nonlinearly, it improves decoding accuracy on simulations and NHP datasets compared to existing methods, showing potential for real-time BCI applications.

**Questions:**

1. Why did the authors only use 20 channels when the dataset provides two 96-channel Utah arrays? Later channels with lower R² can still carry meaningful information.

2. Why is decoding accuracy evaluated only with CC? A CC around 0.6 is not ideal—what is the decoding R²?

3. How is the real-time performance (latency, throughput) of the decoder evaluated in practice?

4. Why does Table 7 use ranks, and what criteria were used for ranking?

5. In Figure 6, what happens beyond 0.8? The trend has not yet plateaued at that point.

6. How difficult is training this network in practice, and what ensures convergence during training?

**Ethical Concerns:**

["NO or VERY MINOR ethics concerns only"]

**Final Justification:**

The authors have added extensive additional experiments, which have resolved my concerns. I am satisfied with the revisions and will increase my score.

**Limitations:**

Yes

**Quality:**

2

**Strengths And Weaknesses:**

Strengths
- Good motivation: tackles multimodal fusion of LFP and spikes, missing data, and timescale mismatch.
- Experiments are detailed and thorough.

Weaknesses
- Overall method novelty is limited: The MRINE method can be roughly understood as a combination of multimodal inputs (LFP and spike signals), nonlinear dynamics to extract latent variables, fusion of these latent variables, and a self-encoding structure to reconstruct neural signals. While this integration is useful, the overall novelty of the method is limited. The approach is largely based on existing techniques, such as multimodal fusion, nonlinear dynamic systems, and autoencoders, without introducing a fundamentally new concept. Additionally, many spike decoding methods bin spikes into time windows and model the firing rates as Gaussian distributions. This method is widely used and effectively captures the high-frequency nature of spike signals. It would be helpful to discuss the advantages and limitations of this approach and compare it with MRINE’s decoding strategy.
- Uses too few channels and sessions: Although the paper claims the dataset contains 96 channels of neural data, the original dataset cited actually includes two 96-channel electrode arrays. However, the experiments in the paper only used 20 channels. This reduced number of channels is somewhat limiting for decoding performance, as even channels with low R² values may still carry valuable movement-related information. If the number of channels increases, a single modality signal (such as spike or LFP) might be sufficient to perform the motion decoding task, which makes the use and fusion of multimodal data seem less necessary. Furthermore, the number of sessions used in the study is limited, which raises concerns about the generalizability of the results. A larger dataset with more sessions would provide a more reliable assessment of the model's robustness and its ability to generalize across different experimental conditions.
- Lacks comparison with some mainstream or latest decoding methods like LFADS, CEBRA, MINT, and BAND.
- Does not consider non-stationarity of neural signals: Neural signals are inherently non-stationary, meaning their statistical properties can change over time. The relationship between neural signals and movement signals is likely time-dependent, and encoding functions may vary over time. However, MRINE assumes that the encoding model remains stationary, which overlooks the time-varying nature of neural data. This assumption could lead to unstable performance in decoding tasks, particularly in long-duration experiments or varying conditions. Incorporating mechanisms to adapt to the non-stationarity of neural signals would make the model more robust and applicable to real-time scenarios.
- Real-time performance is not evaluated: Although the paper claims that MRINE is designed for real-time decoding, it does not provide a detailed evaluation of the model's decoding efficiency. Real-time performance, including latency, throughput, and computational demands, is critical for brain-computer interfaces (BCIs) and other real-time applications.

---

> ### Author Rebuttal · Authors · 2025-07-30
>
> We thank the reviewer for acknowledging the strengths of our problem motivation and experimentation. Below, we address their comments.
>
> > Fundamentally new concepts
>
> This is an important point to clarify. While the basic individual building blocks in MRINE (e.g., MLPs used as part of the encoder) exist, the novelty in MRINE is in how to use these basic blocks to develop a novel overall architecture, training method, and inference method that solves problems of importance to neuroscience and neurotechnology: real-time multimodal inference, different timescales, missing samples. **We clarify that this is similar to many important methods in ML that use existing basic blocks to come up with a new method, for example, all of our baselines are similar in this respect**. We emphasize that the main novelty of MRINE lies in designing a new technique for addressing problems in important applications in neuroscience, rather than proposing new individual deep learning components.
>
> > Gaussian modeling of spikes and additional baselines
>
> This is a great point, and we agree with the reviewer that it’s a natural choice to model binned spikes through Gaussian distribution. Precisely for that reason, we have an ablation study in Appendix A.6.1 that shows that an MRINE variant, which modeled binned spikes through Gaussian distribution, achieved slightly worse but competitive performance compared to that of MRINE. Importantly, through this ablation study, we also show that MRINE’s improved performance does not simply result from modeling two modalities through different observation models (Gaussian vs. Poisson), but also from its model/inference design and training objective.
>
> Also, the above choice would enable testing the performance of single-scale models of neural activity, such as LFADS, on multimodal signals by concatenating them at the input level. Following the reviewer’s recommendation, we now trained LFADS on same timescale spike and LFP signals of NHP grid-reaching dataset. In addition, we also trained CEBRA  models on these multimodal signals by following the CEBRA manuscript authors’ recommendation on multimodal training (i.e., treating each modality as a separate recording session and training a multi-session CEBRA model) (Schneider et al., 2023). As shown below, MRINE outperformed both methods in downstream behavior decoding.
>
> We thank the reviewer for providing this opportunity to provide more evidence, in addition to our existing baseline results across 7 sessions from 2 different datasets on various settings such as different information regimes, different timescale scenarios, and several sample-dropping regimes.
>
> |Model | CC (mean $\pm$ std)|
> |-|-|
> |**MRINE**|	 **0.621 $\pm$ 0.011**|
> |LFADS|	 0.547 $\pm$ 0.011|
> |CEBRA| 0.433 $\pm$ 0.002|
>
> > Using few channels
>
> We thank the reviewer for raising this point. There are two reasons we focused on 20 channels in our original motor dataset. First, the number of channels it takes for motor decoding performance to saturate depends on the task complexity, and this number is typically around or less than 20 neurons for benchmark neural datasets, including the public data we had. For this reason, we went up to 20 neurons because, in that regime, performance was not saturated, so we could ask whether multimodal methods can aggregate information across modalities to improve performance. This saturation is also observed in the prior study that tested MSID. Second, while in laboratory neuroscience experiments, one can record more than 20 neurons, this is not the case for chronic BCIs because they are based on implantable devices with fewer channels, and further because they lose the signal over time due to scar tissue formation. The main advantage of multimodal aggregation would be in low- to mid-information regimes for one modality, such that the other modality can help improve accuracy and robustness, as also shown in prior studies (Bansal et al., 2012; Ahmadipour et al., 2024; Hsieh et al., 2018). For example, prior studies show that spiking activity from implanted electrodes may degrade faster than local field potentials (Stavisky et al., 2015), leading to a small number of spike channels. This is exactly the scenario where using the more robust LFP signals is the most advantageous, as it can provide significant robustness and performance improvements, as we show in the manuscript.
>
> Nevertheless, also per reviewer G325’s request, we now compared MRINE to CEBRA in a 800-D calcium imaging and spike dataset, on which MRINE outperformed CEBRA in downstream visual stimuli decoding task (see item 3 in our response to reviewer G325). Overall, this new analysis shows that MRINE can well generalize to higher-dimensional datasets.
>
> > Real-time compatibility
>
> We thank the reviewer for their comment. We agree with the reviewer that reporting the computational complexity of MRINE is important to ensure its real-time compatibility. Thus, we now report the per-timestep inference times for all methods to give an insight into the computational complexities. MRINE’s latent inference time per timestep is $1.82 \pm 0.13$ ms with Intel i7-10700K processor, which is smaller than the timestep considered in this study (10 ms), and smaller than common brain-computer interfaces (BCI) timesteps (e.g., 50 ms in Stavisky et al., 2015), thus suggesting real-time BCI applicability. Also, the latent inference times per timestep are $1.83 \pm 0.12$ ms for MMGPVAE, $0.03 \pm 0.008$ ms for MSID, and $25.91 \pm 3.44$ ms for mmPLRNN. While closed-loop real-time experiments with animals/humans in the loop constitute a major neurophysiological/experimental effort outside the scope of this modeling work, the above computational times show MRINE’s applicability for such real-time BCI experiments.
>
> > Stationarity of neural signals
>
> We agree with the reviewer that accounting for non-stationarity is important for real-world decoding and discuss this as a limitation in the manuscript. Like many recent models—including MMGPVAE, mmPLRNN, LFADS, and CEBRA—MRINE assumes stationary encoding. Extending MRINE to handle distributional shifts or task-dependent changes is a valuable direction for future work, particularly for long-term or real-time applications. Indeed, developing calibration or alignment methods over time to address time variations is a complementary yet distinct direction of work (Farshchian et al., 2018; Karpowicz et al., 2025).
>
> > Decoding R2
>
> We now provide the 20 Spike - 20 LFP results in Table 1 using R2 below. We show that the relative performance comparisons between MRINE and baseline methods are consistent across CC and R2.
>
>
> NHP grid reaching:
> | Model | Decoding R2 (mean $\pm$ std) |
> | - | - |
> | MVAE | 0.190 $\pm$ 0.009 |
> | MSID | 0.273 $\pm$ 0.013 |
> | mmPLRNN | 0.302 $\pm$ 0.013 |
> | MMGPVAE | 0.334 $\pm$ 0.012 |
> | **MRINE** | **0.375 $\pm$ 0.013** |
>
> NHP center-out reaching:
> | Model | Decoding R2 (mean $\pm$ std) |
> | - | - |
> | MVAE | 0.298 $\pm$ 0.022 |
> | MSID | 0.343 $\pm$ 0.030 |
> | mmPLRNN | 0.294 $\pm$ 0.036 |
> | MMGPVAE | 0.351 $\pm$ 0.029 |
> | **MRINE** | **0.435 $\pm$ 0.030** |
>
> > Using ranks in Table 7
>
> We used ranks in Table 7 because, unlike Fig. 5 (behavior decoding), Fig. 6 (neural reconstruction) did not show a consistent top-performing model across all sample dropping regimes. To summarize overall performance, we ranked the four models (1 = best, 4 = worst) within each regime based on reconstruction accuracy, then averaged these ranks across all regimes. This provides a clearer aggregate comparison when no single model dominates across conditions, as described in Appendix A.4.6.
>
> > Regime beyond 0.8
>
> We limited the sample dropping rate to a maximum of 0.8 because beyond that point, one modality is often almost entirely missing, making it difficult to meaningfully assess multimodal inference. At such extreme sparsity levels, the task shifts from inference to single-modality reconstruction, which is outside the scope of our current evaluation.
>
> > Difficulty of training the model
>
> Training MRINE in practice is not difficult, since we observed that most of the default hyperparameters, such as in Tables 3,4,5 are the same, except for the smoothness regularization parameters for which we recommend a hyperparameter search over a small grid. We used the validation loss as the convergence criterion.
>
> Overall, we thank the reviewer for their careful evaluation and for highlighting several important points, which we have addressed in detail with additional analyses and clarifications. We hope that these responses and the new evidence provided will help the reviewer to reassess their overall evaluation of our work.
>
> References:
>
> - Stavisky et al. (2015). A high performing brain–machine interface driven by low-frequency local field potentials alone and together with spikes. Journal of Neural Engineering.
> - Bansal et al. (2012). Decoding 3D reach and grasp from hybrid signals in motor and premotor cortices: Spikes, multiunit activity, and local field potentials. Journal of Neurophysiology.
> - Ahmadipour et al. (2024). Multimodal subspace identification for modeling discrete-continuous spiking and field potential population activity. Journal of Neural Engineering.
> - Hsieh et al. (2018). Multiscale modeling and decoding algorithms for spike-field activity. Journal of Neural Engineering.
> - Farshchian et al. (2018). Adversarial Domain Adaptation for Stable Brain-Machine Interfaces. International Conference on Learning Representations.
> - Karpowicz et al. (2025). Stabilizing brain-computer interfaces through alignment of latent dynamics. Nature Communications.
> - Schneider et al. (2023). Learnable latent embeddings for joint behavioural and neural analysis. Nature.

---

> > ### Comment · Reviewer_BeqV · 2025-08-05
> >
> > Thank you for the detailed and thoughtful rebuttal. I appreciate the authors' clarifications and the additional experiments, which address most of my concerns. However, I would like to comment on one specific point.
> >
> > In the rebuttal, the authors state:
> >
> > “For this reason, we went up to 20 neurons because, in that regime, performance was not saturated, so we could ask whether multimodal methods can aggregate information across modalities to improve performance.”
> >
> > I respectfully disagree with this reasoning. The motivation for proposing a multimodal fusion approach (spike + LFP) should be to improve performance, especially under realistic and demanding scenarios, rather than only demonstrating improvements in a limited or artificially constrained setting where single-modality performance is still far from saturation. It is expected that adding a second modality helps when the first modality is weak — the more critical question is whether fusion offers additional benefits when one modality already performs strongly.
> >
> > In fact, as I previously noted:
> >
> > If the number of channels increases, a single modality signal (such as spike or LFP) might be sufficient to perform the motion decoding task, which makes the use and fusion of multimodal data seem less necessary.
> >
> > I also remain curious about the trend beyond the 0.8 sample-dropping regime in Figure 6. Seeing the performance of a single remaining modality would provide valuable insights into the limits and necessity of multimodal fusion, especially under asymmetric modality loss.
> >
> > Additionally, while the authors argue that 20 channels reflect real-world constraints in chronic BCIs, in practice, many chronic BCI systems use 96 or more channels, and even if only one-third of them yield high-quality signals, that still exceeds 20. Furthermore, the dataset used in this paper contains significantly more than 20 channels, so I would much prefer to see results with more channels from the original dataset, rather than results on a different dataset (such as the 800-D calcium imaging dataset). The current setting still feels overly restricted.
> >
> > For these reasons, I will maintain my original score.

---

> ### Author Response · Authors · 2025-08-08
>
> > Number of channels
>
> We thank the reviewer for raising these great points. To demonstrate MRINE’s performance beyond 20-20 channel regime per reviewer’s request, we now trained MRINE, single-scale spike and LFP models, MSID, and MMGPVAE in 30-30 and 60-60 spike and LFP channel regimes.
>
> |Model|30 Spike - 30 LFP|60 Spike - 60 LFP|
> |-|-|-|
> |Single-scale LFP|0.478|0.456|
> |Single-scale Spike|0.618|0.637|
> |MSID|0.573|0.597|
> |MMGPVAE|0.516|0.542|
> |**MRINE**|**0.620**|**0.693**|
>
> As shown in the table above, **MRINE outperformed all single-scale and multimodal baselines in these high-channel regimes**. This result shows **that MRINE is also advantageous in the high-channel count regime in both the spike-LFP motor dataset as well as the new calcium/neuropixel dataset**. We sincerely thank the reviewer for their great suggestion.
>
> That being said, below we provide reasons why performance comparisons in the 20-channel regime are also valuable for future BCI research:
>
> 1. Viable recording channels drop over time post-implantation: While we agree that some chronic BCI systems deploy 96-channel arrays, the number of reliably recording channels post-implantation is often much lower. For example, the BrainGate study (Harris and Tyler, 2013) reported that the number of active spike units can drop to approximately 24 units over time. Similarly, Barrese et al. (2013) identified several electrode failure modes and found that the percentage of viable recording channels can fall to as low as 5–24%. These observations reflect real-world constraints that show the importance of lower-channel regimes.
>
> 2. Spike signals degrade faster than LFPs: Beyond hardware degradation above, signal stability differs between modalities. Prior work has shown that LFP signals exhibit significantly greater stability over time compared to spikes, with spikes degrading faster (Flint et al., 2016; Wang et al., 2014; Heldman and Moran, 2020; Sharma et al., 2015). This can result in a substantial reduction in usable spike channels over time. In such scenarios, robust multimodal methods like MRINE can be particularly valuable by aggregating information from a limited number of spike channels with the more abundant, stable LFP signals (please note the performance difference between single-scale LFP model and MRINE). Indeed, multimodal methods can make a big difference to implanted patients in these low-channel regimes: **they allow these patients to keep using their implanted BCI for longer**.
>
> 3. Some devices can have lower channel count: Recent developments in wireless BCI systems have explored lower channel count designs to reduce bandwidth and power demands. For example, Uran et al. (2022) introduced a neural SoC system using 16 channels with on-chip feature extraction, illustrating that such configurations remain relevant in certain application contexts.
>
> > Drop beyond 0.8
>
> Per reviewer’s request, for the NHP grid-reaching dataset and for MRINE, MMGPVAE, and MSID, we computed the neural reconstructions in Fig. 6 also when the modality of interest is completely missing but reconstructed from the other modality. In this scenario, reconstruction quality significantly decreased for all models, for instance, MRINE, MSID and MMGPVAE achieved 0.592, 0.584, and 0.571 AUC for spikes and 0.244, 0.244, and 0.041 CC for LFP signals. We believe that this is expected since neither of these models’ training objectives is cross-modal reconstruction, but rather on self-prediction or self-reconstruction by aggregating information. To achieve competitive cross-modal reconstruction, the training objective should be modified accordingly, and we will include this limitation (that is not only inherent to MRINE) in the discussion paragraph.
>
> References:
>
> - Harris and Tyler (2013). Biological, Mechanical, and Technological Considerations Affecting the Longevity of Intracortical Electrode Recordings. Critical Reviews in Biomedical Engineering.
>
> - Barrese et al. (2013). Failure mode analysis of silicon-based intracortical microelectrode arrays in non-human primates. JNE.
>
> - Flint et al. (2016). Long-Term Stability of Motor Cortical Activity: Implications for Brain Machine Interfaces and Optimal Feedback Control. JNE.
>
> - Sharma et al. (2015). Time Stability and Coherence Analysis of Multiunit, Single-Unit and Local Field Potential Neuronal Signals in Chronically Implanted Brain Electrodes. Bioelectronic Medicine.
>
> - Heldman and Moran (2020). Chapter 20—Local field potentials for BCI control. In N. F. Ramsey & J. del R. Millán (Eds.), Handbook of Clinical Neurology (Vol. 168, pp. 279–288). Elsevier.
>
> - Wang et al. (2014). Long-term decoding stability of local field potentials from silicon arrays in primate motor cortex during a 2D center out task. JNE.
>
> - Uran et al. (2022). A 16-Channel Neural Recording System-on-Chip With CHT Feature Extraction Processor in 65-nm CMOS. IEEE Journal of Solid-State Circuits.

---

### Official Review · Reviewer_G325 · 2025-06-29

**Clarity:** 4
**Significance:** 3
**Originality:** 3
**Rating:** 4
**Confidence:** 4

**Summary:**

The paper proposes a method based on dynamical systems for analyzing multi-modal, multi-scale neural data. The proposed method, unlike prior methods, accounts for differences in scales (sampling rates) in the modalities. It first encodes each modality separately and applies non-linear fusion to learn a shared representation.  From this representation, it reconstructs the data across modalities.  Through shared representation, it allows for missing data. Using both synthetic and real-world experiments, the paper demonstrates improved behavioral decoding over both single-modal model and multi-modal models, which do not account for timescale differences.

**Questions:**

Questions on architecture

* On the experiments, including results with more common evaluation metrics such as R2, and considering realistic data settings (see weakness W1) could improve the paper.

* In Figure 1b, the indexing for both modalities appears to be the same. Do you apply prediction for a common set to time indices, before concatenation?

* If Fig 1b, And are the MLPs time invariant (the arrows indicate all inputs (across time) are fed to it at once)?  Maybe adding clarifying text could help.

*  On the smoothness regularizers (Eqn. 10, 3rd term), the choice for fast/slow latent dimensions seems arbitrary. Any insights on the effect of this choice?

**Ethical Concerns:**

["NO or VERY MINOR ethics concerns only"]

**Final Justification:**

The paper deals with a well-motivated problem. My main concern was with the experiment setup. The authors used a small number of channels in the experiments, which does not fit very well with realistic neural datasets. This limits the application of the method in real datasets. While the authors have provided some discussion during rebuttal, my concern is not fully addressed. Therefore, I maintain my original score.

**Limitations:**

yes

**Quality:**

3

**Strengths And Weaknesses:**

**Strength**


S1: The paper is well-written, with a good literature review and a clear problem statement. The paper provides a good argument for why the non-causal methods fail in multi-scale settings, motivating a causal approach.



S2: While other methods have proposed using deep learning based dynamical models for single-scale and also multi-scale linear LDS models, this paper considers a newer challenge (at least for deep dynamical models) of non-linearly fusing information across modalities, while still maintaining real-time inference, and handling missing data.



S3: The work conducted an interesting set of experiments to show the utility of fusing information across multiple modalities, while still accounting for their timescale differences of the modalities.  The ablation studies demonstrate the effectiveness of different components of the model in improving behavioral decoding in a multi-scale setting.

Weakness


W1: The number of channels used for the experiment (5-20) is rather small. This deviates from a realistic setting with recordings having 100s of simultaneously recorded neurons (channels). The plots shown in Figure 3 show the diminishing returns in about 10 channels. The current experiments are limited in assessing the impact of the method in more realistic settings.


W2: The evaluation metric for both latent reconstruction (section 4.1) and decoding accuracy (section 4.2) somewhat feels unjustified, since most prior approaches use R2 to directly measure the reconstruction error instead of correlation. How does the correlation inform us about the performance of the model?  Therefore, interpreting percentage improvement in this metric is hard. And plots comparing groundtruth and inferred latents/behavioral quantities could be helpful to see.


W3: The experiments appear to be limited to LFP and spikes, but cited prior methods like MMGPVAE explore wider ranges of multi-modal data, such as (temporal) video recordings of behavior, calcium imaging etc. Absence of other modalities beyond LFP and spikes limits the broader application of the method.

---

> ### Author Rebuttal · Authors · 2025-07-30
>
> We thank the reviewer for providing an excellent summary of our work and its strength. Below, we address their comments:
>
> > Small number of channels
>
> This is a great insight, we thank the reviewer for raising that. There are two reasons why we focused on 20 channels in our original motor dataset. First, the number of channels it takes for motor decoding performance to saturate depends on the task complexity, and this number is typically around or less than 20 neurons for benchmark neural datasets, including the public data we had. For this reason, we went up to 20 neurons because, in that regime, performance was not saturated, so we could ask whether multimodal methods can aggregate information across modalities to improve performance. This saturation is also observed in the prior study that tested MSID. Second, while in laboratory neuroscience experiments, one can record more than 20 neurons, this is not the case for chronic BCIs because they are based on implantable devices with fewer channels, and further, because they lose the signal over time due to scar tissue formation. The main advantage of multimodal aggregation would be in low- to mid-information regimes for one modality, such that the other modality can help improve accuracy and robustness, as also shown in prior studies (Bansal et al., 2012; Ahmadipour et al., 2024; Hsieh et al., 2018). For example, prior studies show that spiking activity from implanted electrodes may degrade faster than LFPs (Stavisky et al., 2015), leading to a small number of spike channels. This is exactly the scenario where using the more robust LFP signals is the most advantageous, as it can provide significant robustness and performance improvements, as we show in the manuscript.
>
> Nevertheless, to demonstrate that MRINE is not limited to low-dimensional inputs, we additionally evaluated it on a high-dimensional (800-channel) calcium imaging and spike dataset (see item 3 below). MRINE outperformed CEBRA and all other baselines on this task, supporting its scalability to more realistic, large-scale settings.
>
> > Decoding performance on R2:
>
> Per the reviewer’s request, we now provide the 20 Spike - 20 LFP results in Table 1 using R2 below. We show that the relative performance comparisons between MRINE and baseline methods are consistent across CC and R2.
>
> NHP grid reaching:
> | Model | Decoding R2 (mean $\pm$ std) |
> | - | - |
> | MVAE | 0.190 $\pm$ 0.009 |
> | MSID | 0.273 $\pm$ 0.013 |
> | mmPLRNN | 0.302 $\pm$ 0.013 |
> | MMGPVAE | 0.334 $\pm$ 0.012 |
> | **MRINE** | **0.375 $\pm$ 0.013** |
>
> NHP center-out reaching:
> | Model | Decoding R2 (mean $\pm$ std) |
> | - | - |
> | MVAE | 0.298 $\pm$ 0.022 |
> | MSID | 0.343 $\pm$ 0.030 |
> | mmPLRNN | 0.294 $\pm$ 0.036 |
> | MMGPVAE | 0.351 $\pm$ 0.029 |
> | **MRINE** | **0.435 $\pm$ 0.030** |
>
> We will also include example decoded trajectories in our manuscript.
>
> > Broader application of the method
>
> We thank the reviewer for their suggestion. We now tested MRINE on a completely new dataset that contained high-dimensional calcium imaging and spike data, where the downstream task is to predict the visual stimuli (ID of the frame) as done in CEBRA (Schneider et al. 2023). We compare the mean absolute errors (MAE) of frame ID prediction for MRINE, CEBRA, single-scale (SS) models trained on either calcium imaging or spike data, and for decoding directly from the data modalities using naive Bayes classifiers. MRINE outperforms all baselines:
>
> | Model | MAE|
> |-|-|
> |Calcium imaging| 151.60|
> |Spike|67.76|
> |SS calcium imaging|22.66|
> |SS spike|10.94|
> |CEBRA|9.31|
> |**MRINE**|**5.09**|
>
> > Indices in Fig. 1b
>
> We used the same time index for latent representations of both modalities as modality-specific LDM of the slower timescale signals extracts latent representations in the timescale of the faster signal (i.e., spikes). We explain the procedure in L188-203.
>
> > MLP in Fig. 1b
>
> The reviewer’s understanding is indeed correct, MLP modules are time invariant, and the dynamical information is leveraged through linear dynamical models. We will make sure to add text in the figure to make this clear.
>
> > Slow/fast scales on smoothness regularization
>
> This is a great question. While we don’t report full results in the paper, we observed during development that applying smoothness regularization to half of the latent dimensions yielded equal or better performance than applying it to all dimensions. Indeed, this avoids overly strong regularization, which can bias the model toward overly smooth dynamics and suppress informative but irregular neural patterns.
>
> References:
> - Stavisky, S. D., Kao, J. C., Nuyujukian, P., Ryu, S. I., & Shenoy, K. V. (2015). A high performing brain–machine interface driven by low-frequency local field potentials alone and together with spikes. Journal of Neural Engineering.
> - Bansal, A. K., Truccolo, W., Vargas-Irwin, C. E., & Donoghue, J. P. (2012). Decoding 3D reach and grasp from hybrid signals in motor and premotor cortices: Spikes, multiunit activity, and local field potentials. Journal of Neurophysiology.
> - Ahmadipour, P., Sani, O. G., Pesaran, B., & Shanechi, M. M. (2024). Multimodal subspace identification for modeling discrete-continuous spiking and field potential population activity. Journal of Neural Engineering.
> - Hsieh, H.-L., Wong, Y. T., Pesaran, B., & Shanechi, M. M. (2018). Multiscale modeling and decoding algorithms for spike-field activity. Journal of Neural Engineering.
> - Schneider, S., Lee, J. H., & Mathis, M. W. (2023). Learnable latent embeddings for joint behavioural and neural analysis. Nature.

---

> ### Comment · Reviewer_G325 · 2025-08-02
>
> I thank the authors for answering my questions and doing additional experiments on calcium data.
>
> > For this reason, we went up to 20 neurons because, in that regime, performance was not saturated, so we could ask whether multimodal methods can aggregate information across modalities to improve performance. This saturation is also observed in the prior study that tested MSID.
>
> >Per the reviewer’s request, we now provide the 20 Spike - 20 LFP results in Table 1 using R2 below.
>
> I thank the authors for being candid about these limitations. However, I still think the choice of dataset may not be appropriate for your model and does not reflect the realistic settings. Therefore, I will keep my score.

---

> > ### Author Response · Authors · 2025-08-03
> >
> > We thank the reviewer for their continued engagement and for acknowledging our additional experiments. Regarding datasets and evaluation, we would like to clarify three key points.
> >
> > First, to demonstrate the scalability of MRINE beyond low-channel motor cortical datasets, we evaluated it on a high-dimensional (800-channel) calcium imaging and Neuropixel spike dataset. This dataset includes substantially more units (100’s) and reflects a realistic large-scale recording scenario. We showed that again in this case, MRINE outperforms the baseline methods, including CEBRA. This new dataset evaluation shows that a large number of units can be incorporated into MRINE, and this is not a limitation of MRINE.
> >
> > Second, with this new calcium dataset in addition to the prior 2 distinct motor datasets, we have now shown the advantage of MRINE across 3 diverse real-world datasets with: 1) different channel counts, 2) different modalities—spikes, LFP, calcium imaging, Neuropixels—, 3) in both motor and visual cortical areas, 4) for decoding two distinct motor tasks and a visual task, and 5) in both mice and monkeys. These datasets thus cover diverse real-world setups for our model evaluations.
> >
> > Third, all methods considered in this work, including recent SOTA multimodal neural data models such as MMGPVAE and mmPLRN, were trained and evaluated on the exact same datasets. The fact that MRINE consistently outperforms these baselines across multiple datasets and tasks highlights the strength and generality of the approach.
> >
> > We hope that the above clarifies that our findings are not dependent on dataset choice, but rather reflect improvements enabled by MRINE across 3 diverse datasets, tasks, modalities, and brain regions.

---

> > ### Author Response · Authors · 2025-08-09
> >
> > Once again, we thank the reviewer for their feedback and suggestions. As a follow-up to their earlier comment, we have now trained MRINE in a higher-channel regime in the original spike-LFP motor dataset. In this setting, MRINE trained on 60 spike and 60 LFP channels achieved a 0.693 decoding CC, outperforming the single-scale spike (0.637 CC) and LFP (0.456 CC) baselines, as well as multimodal MSID (0.597 CC) and MMGPVAE (0.542 CC), as also presented in our final response to reviewer BeqV. These results, combined with our results on the high-channel calcium/neuropixel data, confirm that the advantage of MRINE also extends to the high-channel case.

---

### Official Review · Reviewer_FWMt · 2025-07-02

**Clarity:** 2
**Significance:** 2
**Originality:** 2
**Rating:** 4
**Confidence:** 3

**Summary:**

The submission considers real-time bi-modal inference of a discrete and continuous neural modality evolving at different timescales with missing data. The authors combine several ideas in their work, including a multi-scale encoder and recursive variational inference based on Kalman filtering or smoothing. Empirical results on NHP indicate that the method improves the decoding and reconstruction performance compared to previous approaches.

**Questions:**

To clarify, (5) means that $a_t$ is deterministic given $s_t$ and $y_t$, so these factors have a point mass distribution. But then in (6) and (7), the modality-specific versions seem to be stochastic with Gaussian noise, and how they are related is not fully clear to me. As far as I understand $x_{t|t}^s$ is then just the mean of the filtering distribution of the latent factor, and does not capture the variance/covariances of the different factor dimensions?

Is T’ subset T a necessary constraint, so that the discrete modality evolves faster?

In the model description 3.2, it is not clear where the last MLP from Fig.1 comes in?

It would be useful to analyse how previous work perform in the stochastic Lorenz attractor simulations. Does the relative performance of the suggested approach compared to previous work in this example depend on the specific task/evaluation, such as latent reconstruction accuracy, real-time k-step ahead data predictions? How sensitive are these evaluations with respect to the weighting hyperparameters in the loss function?

Why do the authors suggest the smoothness regularization term, instead of just using a KL penalty for sequential VAEs? Do the smoothness parameters adapt to the scale of each modality?

**Ethical Concerns:**

["NO or VERY MINOR ethics concerns only"]

**Final Justification:**

The detailed response that has clarified most points (motivation for the multiscale embedding factor, details on the computational complexity of their approach in comparison to previous works, a new benchmark dataset, imputation approaches used in the baseline method). I will therefore increase my score to a weak accept.

**Limitations:**

Yes

**Quality:**

3

**Strengths And Weaknesses:**

The combination of different regularisers, model architectures and inference models is indeed interesting.

The zero-imputation for missing values for the baseline methods seems a bit simple. It would be interesting to see how alternative approaches would peform using less naïve imputations, e.g. just a simple Kalman filter for the continuous modality.

The rationale why one needs the ‘multiscale embedding factors’ a are not fully clear to me. Why can I not just use x instead (and pre-condition with C, and incorporate the noise r into the noise w?).

The computational complexity of the method relative to the baselines should be made clearer. For example, I would assume that the method can be beneficial compared with the multi-modal GP approach which may require some approximations for many observations.

I found the presentation to be a bit unclear at times. For example, how are the factors and parameters in (7) with ^s and ^y related to the counterparts in (1)? Does this mean that in the generative path (1), there is no independence assumption, while on the inference/encoder path, the two modalities are encoded independently until they are aggregated via an MLP at the final layer?

---

> ### Author Rebuttal · Authors · 2025-07-30
>
> We thank the reviewer for their questions and comments. Please see our answers below:
>
> > Meaning and rationale of ‘multiscale embedding factors’ $a_t$
>
> The reviewer is correct that $a_t$ is deterministically computed from the input signals ($s_t$, $y_t$) via the multiscale encoder. Indeed, $a_t$ has a point mass conditioned on the inputs. This holds for both the multimodal case in Eq. (5) and the modality-specific versions in Eqs. (6) and (7).
>
> After obtaining $a_t^s$ and $a_t^y$ through the modality-specific MLPs in Fig. 1b, they serve as the observations for modality-specific LDMs. As such, we apply Kalman filtering on modality-specific LDMs to account for timescale differences and obtain $a_{t|t}^s$ and $a_{t|t}^y$.
>
> Although $a_t, a_t^s$, and $a_t^y$ are deterministically obtained from neural data, the latent factors $x_t, x_t^s$, and $x_t^y$ in multiscale and modality-specific LDMs are stochastic (conditioned on $a_t, a_t^s$, and $a_t^y$) due to these LDMs’ process and observation noise parameters, which are learned during training. We apply Kalman filtering to these LDMs, yielding $x_{t|t}$ as the posterior mean of the latent state. This filtered estimate is then linearly projected to obtain $a_{t|t} = C x_{t|t}$ (and similarly, $a_{t|t}^s = C_s x_{t|t}^s$ for spikes and $a_{t|t}^y = C_y x_{t|t}^y$ for LFP, which are later fused through the last MLP in Fig. 1b to form $a_t$). While $a_{t|t}$ is also a point estimate (mean of posterior), it is a denoised and temporally smoothed version of the raw $a_t$, benefiting from the learned temporal structure in the LDMs that enable Kalman filtering/denoising over time. Also, the Kalman filter computes this point estimate based on the trained process and observation noise covariances, which together dictate the Kalman gain and posterior covariance. In this way, although the posterior covariance is not explicitly used in our loss, the Kalman filtering process implicitly captures temporal uncertainty and regularity through the explicit use of process and observation noise covariances, which the posterior covariance is a function of.
>
> Further, the reason for having $a_t$ in our modeling instead of using a single latent factor ($x_t$) is to form an LDM so that we can enable Kalman filtering. Specifically, here $a_t$ becomes the observations of the LDM, which relates it to $x_t$. Through Kalman filtering, we can effectively leverage the learned inherent dynamics for imputing intermediary/missing timesteps, and allow for both real-time recursive filtering and non-causal smoothing that provides further flexibility during inference time. If we directly connect $x_t$ to $y_t$ and $s_t$ without $a_t$, no LDM is formed and thus Kalman filtering cannot be utilized to achieve the above capabilities.
>
> > Imputation of missing samples
>
> We definitely agree with the reviewer. This is exactly why we included MSID in our baselines, as it handles missing samples through Kalman filtering for continuous modality, and point process filtering for the discrete modality. We show that the MRINE outperforms MSID. For the deep-learning architectures, their model architectures do not support such a setting, so we applied zero-imputation as is common practice.
>
> While simple, this approach has been found effective in several domains. For instance, in a recent foundational modeling study on wearable behavioral data, zero-imputation of input signals outperformed learnable interpolation and tokenization-based alternatives designed to circumvent imputation entirely (Erturk et al., 2025).
>
> > Computational complexity
>
> We thank the reviewer for their comment. We now report the per-timestep inference times for all methods to give an insight into the computational complexities. MRINE’s latent inference time per timestep is $1.82 \pm 0.13$ ms with Intel i7-10700K processor, which is smaller than the timestep considered in this study (10 ms), and smaller than common brain-computer interface (BCI) timesteps (e.g., 50 ms in Stavisky et al., 2015), thus suggesting real-time BCI applicability. Also, the latent inference times per timestep are $1.83 \pm 0.12$ ms for MMGPVAE, $0.03 \pm 0.008$ ms for MSID, and $25.91 \pm 3.44$ ms for mmPLRNN. The multi-modal GP approach had a similar inference time compared to MRINE, as it does not require time-recurrence but processes the whole data as a batch (which limits its real-time capabilities, as inferring latent factors requires using the future timesteps of neural data).
>
> > Independence assumption
>
> During the generation path, we make a conditional independence assumption that $p(y_t, s_t | a_t) = p(y_t | a_t) p(s_t | a_t)$. Multiscale embedding factors $a_t$’s are a function of their modality-specific counterparts, i.e., $a_t = h(a_t^s, a_t^y)$ where $h(\cdot)$ is composed of modality-specific LDMs and the final fusion layer ($\phi_m$). As our goal is to extract latent embeddings by fusing information across both modalities, as the reviewer stated, we do not make independence assumptions on the inference/encoder path similar to product-of-experts, or enforce disentanglement across modality-specific factors (factors being merged at the final MLP) through a loss term. We believe that incorporating such independence assumptions and disentanglement are some great future directions to investigate for a better performance.
>
> > $T$ and $T’$
>
> This is a great question. We indeed assume that $T’$ is a subset of $T$, indicating that the discrete modality evolves faster, in line with the prior work (spikes have a faster sample rate than LFPs). However, our model formulation can be extended to scenarios where $T’$ is not a subset of $T$. In those scenarios, we would represent both signals in the greatest-common-divisor timescale with appropriate masks to indicate the availability of the signal samples. For instance, if spike and LFP signals were sampled at 20 and 50 ms, we could represent each signal at 10ms resolution during modeling, unmasking the spikes every other timestep and LFPs every 5 steps.
>
> > Last MLP in Fig. 1
>
> The last MLP in Fig. 1b is the fusion module ($\phi_m$) that fuses the $a_{t|t}^y$ and $a_{t|t}^s$ to form $a_t$, as we explained in L200.
>
> > Comparisons in other tasks
>
> Thank you for the suggestion. The Lorenz simulations were designed primarily to validate MRINE’s information aggregation capabilities in a controlled setting, rather than to benchmark against prior work. We believe real neural datasets offer a more meaningful testbed, and accordingly, we evaluate all methods across two distinct datasets with different behavioral tasks and subjects, and across different timescale regimes, multiple information levels, and several sample-dropping scenarios. To further support MRINE’s generalization, we now include new results on a third dataset with another distinct task (see item 3 in our response to reviewer G325), where MRINE again outperforms CEBRA and single-scale models.
>
> As shown in Appendix A.6, hyperparameters—particularly smoothness regularization—do impact performance, and removing them narrows the gap to baselines. This shows the benefit of these regularization components. Incorporating automated hyperparameter tuning (e.g., population-based training) is a promising direction for future work.
>
> > KL divergence term as in SAEs
>
> Thank you for the insightful question. The smoothness regularization terms in our model are implemented as KL divergences between latent representations at consecutive timesteps. The corresponding weights are chosen to reflect the scale of each modality.
>
> During our development phase, we initially explored ELBO-based training, as in sequential VAEs (e.g., SAEs), but found it to be unstable and consistently underperforming in our setting. Our current objective provided more stable optimization and better decoding performance.
>
> To further support this design choice, we now include comparisons with LFADS (an SAE variant) in item 2 of our response to reviewer BeqV, where MRINE demonstrates stronger performance.
>
> Overall, we hope that the above analyses, which provide further evidence on our model’s performance and generalization performance, address the reviewer’s concerns and encourage the reviewer to reconsider our work.
>
> References:
>
> - Erturk, E., Kamran, F., Abbaspourazad, S., Jewell, S., Sharma, H., Li, Y., Williamson, S., Foti, N. J., & Futoma, J. (2025). Beyond Sensor Data: Foundation Models of Behavioral Data from Wearables Improve Health Predictions. Forty-second International Conference on Machine Learning.
> - Stavisky, S. D., Kao, J. C., Nuyujukian, P., Ryu, S. I., & Shenoy, K. V. (2015). A high performing brain–machine interface driven by low-frequency local field potentials alone and together with spikes. Journal of Neural Engineering.

---

> > ### Comment · Reviewer_FWMt · 2025-08-06
> >
> > I thank the authors for their detailed response that has clarified most points. In particular, the motivation for the multiscale embedding factor has become clearer. The authors also provided details on the computational complexity of their approach in comparison to previous works, as well as a new benchmark dataset. The authors also clarified the imputation approaches used in the baseline methods. I will therefore increase my score to a weak accept.

---

### Official Review · Reviewer_VG94 · 2025-07-03

**Clarity:** 3
**Significance:** 3
**Originality:** 3
**Rating:** 5
**Confidence:** 3

**Summary:**

The paper introduces a multi-modal encoder/decoder architecture that is specially designed to work with modalities that have different timescales (e.g. binned spikes, and LFP signals). The architecture is also robust to missing samples in either modality. Empirical evaluation is performed on synthetic, as well as real-world (offline) decoding datasets.

**Questions:**

- What does “primary” modality mean? From the description of MRINE, i don’t see any reason to consider one of the 2 modalities as primary. Does “primary” mean the modality with the fastest timescale? If this is the case, then how can Gaussian modality be primary?

- In Fig 2a and 3a, the points corresponding to number of LFP channels = 0 -- Are these results from single-modality and single-timescale versions of MRINE?

**Ethical Concerns:**

["NO or VERY MINOR ethics concerns only"]

**Final Justification:**

I think the paper is good. My questions were mostly conceptual and the authors have correctly answered them. I was confused between giving a 4 or a 5 rating. But leaving a review at a "borderline" recommendation doesn't seem good, so chose 5.

**Limitations:**

Yes

**Quality:**

4

**Strengths And Weaknesses:**

## Strengths:

- Statistical significance tests have been performed for most results. This is highly appreciated!

- The description of the method is relatively clear.

- Excellent ablation studies presented in the Appendix.


## Weaknesses:

- The method introduces a few hyperparameters. A description for how these were tuned would be beneficial for further use and adaption of the presented method.

---

> ### Author Rebuttal · Authors · 2025-07-30
>
> We thank the reviewer for their thoughtful feedback and for highlighting these important clarifications. Below, we address the reviewer’s comments:
>
> > Hyperparameters
>
> We thank the reviewer for their comment. The HPs of MRINE, except time-dropout and smoothness regularization scales, are typical HPs in any deep-learning application, and we followed educated/common choices. For the smoothness regularization scales, the HP search on each dataset is performed using a random inner-training and inner-validation split from the training set of the first fold on the first available session over a small grid. We will make sure to include these details in the manuscript.
>
> > Meaning of primary
>
> We apologize for the confusion caused by our use of the term “primary.” In our manuscript, we use “primary” to refer to the modality whose dimensionality is held fixed during scaling experiments, while the other modality is gradually added. Importantly, the designation of “primary” is symmetric and does not imply that one modality is inherently more important. For instance, in Figs. 3a and 3e, spike channels are treated as the “primary” modality because their dimensionality is fixed while the number of LFP channels increases. Conversely, in Figs. 3b and 3f, LFP is considered the “primary” modality. This framing was intended to demonstrate that performance improvements are bidirectional: adding either modality to the other enhances decoding performance.
>
> > About Figs 2a and 3a
>
> The reviewer’s interpretation is correct. The points corresponding to 0 LFP channels (or 0 spike channels) represent the single-modality versions of MRINE. These serve as our unimodal baselines and reflect models trained using only one signal modality and its corresponding timescale.

---

> > ### Comment · Reviewer_VG94 · 2025-08-03
> > **Rebuttal Response**
> >
> > I thank the authors for answering my questions and concerns in their rebuttal. I am satisfied with the response, and will maintain my score.

---

### Note · Authors · 2025-08-12

Dear Reviewers and Area Chair,

We are very grateful for all the constructive feedback, which has substantially strengthened our manuscript.

During the rebuttal and discussion period, we included several new analyses following the reviewer suggestions, including:

- **Providing comparisons to additional baselines**, namely LFADS and CEBRA, and showing that MRINE also outperforms them.
- **Adding a new high-channel (i.e., 800-dimensional) visual-stimuli dataset** containing Neuropixel spike and calcium data, and showing that MRINE outperforms single-scale models and CEBRA in the frame prediction task. This also shows MRINE’s generalizability to distinct modalities (calcium + Neuropixel, spikes + LFP), tasks (frame detection, movement decoding), and brain regions (visual, motor).
- **Providing R2 decoding performance** in addition to the Pearson’s correlation coefficient (CC), showing that our performance comparisons remain consistent across different metrics.
- **Computational complexity details**, indicating MRINE’s suitability for real-time settings.
- **Providing higher channel count results for the NHP spike-LFP motor dataset**, showing that MRINE again outperformed baselines and successfully achieved multimodal information aggregation in this high-channel count regime as well. Together with our other results, this shows MRINE’s applicability across different regimes of relevance to chronic BCIs.

We thank the reviewers again for their helpful comments. We believe that these new results and our detailed responses address these concerns and further underscore the strength of MRINE. Thank you again for all your thoughtful feedback and engagement, which has improved our manuscript.

Authors

---

### Decision · Program_Chairs · 2025-09-17

**Decision:**

Accept (poster)

**Comment:**

In this manuscript titled "Dynamical modeling of nonlinear latent factors in multiscale neural activity with real-time inference", the authors proposed a multi-modal multi-timescale real-time decoding network architecture and loss function for brain-computer interface applications. The method consists of Linear Dynamical Models to account for temporal integration, timescale differences, and predict missing data, then performs nonlinear information fusion via a fusion network. It represents an original solution to an important engineering application that of some interest to the NeurIPS community. The conceptual novelty is in the combination of well-known building blocks and the experiments were somewhat unrealisitically used small number of channels, limiting its application in more realisitic scenario where array recordings are typically much higher dimension these days. However, the authors were able to provide very high-dimensional experiments during the rebuttal. I strongly encourage the authors to produce a significantly updated manuscript that includes the details of the additional experiments.